# Characterizing barren plateaus in quantum ansätze with the adjoint representation

Enrico Fontana [1,2], Dylan Herman[1] ✉, Shouvanik Chakrabarti[1], Niraj Kumar[1], Romina Yalovetzky[1], Jamie Heredge[1,3], Shree Hari Sureshbabu[1] & Marco Pistoia [1]

Variational quantum algorithms, a popular heuristic for near-term quantum computers, utilize parameterized quantum circuits which naturally express Lie groups. It has been postulated that many properties of variational quantum algorithms can be understood by studying their corresponding groups, chief among them the presence of vanishing gradients or barren plateaus, but a theoretical derivation has been lacking. Using tools from the representation theory of compact Lie groups, we formulate a theory of barren plateaus for parameterized quantum circuits whose observables lie in their dynamical Lie algebra, covering a large variety of commonly used ansätze such as the Hamiltonian Variational Ansatz, Quantum Alternating Operator Ansatz, and many equivariant quantum neural networks. Our theory provides, for the first time, the ability to compute the exact variance of the gradient of the cost function of the quantum compound ansatz, under mixing conditions that we prove are commonplace.

Variational quantum algorithms (VQAs) are a popular class of quantum computing heuristics due to their low circuit cost and ability to be trained in a hybrid quantum-classical fashion[1]. The community has identified a variety of potential applications for VQAs in the areas of optimization[2–7] and machine learning[8–12]. Unfortunately, the optimization of VQAs can be a computationally challenging task due to (1) exponentially many parameters being required to ensure convergence[13–17], and (2) exponentially many samples being required to estimate gradients, known as the *barren plateau* (BP) problem[18–21]. In some cases, it has been observed numerically that both of these obstacles to VQA optimization can be mitigated when the chosen parameterized quantum circuit (PQC) obeys certain symmetries[14,22]. The symmetries of the ansatz cause its action, in either the Schrödinger or Heisenberg pictures, to break into invariant subspaces. However, there have only been a few cases in which potentially useful symmetries, mostly in the Schrödinger picture, have been identified for analyzing BPs, e.g. permutation invariant subspaces[23]. Previously, symmetries have been leveraged for efficient classical simulation of quantum circuits in both the Schrödinger[24] and Heisenberg[25–27] pictures. The simulation is performed separately in each invariant subspace defined by the symmetries by projecting the states or operators accordingly.

The existing theoretical results on the trainability and convergence of ansätze with symmetries have been restricted to the Schrödinger picture and a setting called *subspace controllable*[14,18,22,23]. Subspace controllability occurs when the circuit can express any unitary transformation between states in an invariant subspace, and it has been observed that it results in training landscapes that are essentially trap-free[28,29]. In addition, if the invariant subspaces have small dimension, i.e., scale polynomially in system size, it can be easily shown that BPs are not present for subspace-controllable PQCs.

These results, however, fail in the uncontrollable setting, where the circuit is limited to expressing a subgroup of the unitary group in the invariant subspace. With respect to the BPs problem, existing work has observed a desirable feature of subspace uncontrollable circuits[22]. In this setting, it appears that the trainability of the ansatz depends on the dimension of the *dynamical Lie algebra* (DLA), which holds almost trivially in the subspace-controllable setting since the DLA dimension grows with the square of the subspace dimension. However, existing work has only provided evidence of this connection to the DLA dimension numerically in the uncontrollable setting[22]. There are cases

[1]Global Technology Applied Research, JPMorganChase, New York, NY 10017, USA. [2]Computer and Information Sciences, University of Strathclyde, Glasgow G1 1XQ, UK. [3]School of Physics, The University of Melbourne, Parkville, VIC 3052, Australia. ✉e-mail: dylan.a.herman@jpmchase.com

where, for uncontrollable PQCs, the dimension of the effective DLA only grows polynomially in the system size, while the invariant subspace dimension where the initial state lies is exponentially growing, such as the quantum compound ansatz[30,31]. Note that the effective DLA is the restriction of the action of the DLA to an invariant subspace. Thus, this connection between the DLA dimension and BPs has remained unproven in the general setting.

In this work, using a simple but powerful observation regarding the adjoint representation and the representation theory of compact Lie groups, we prove that for a general class of PQCs the variance of the gradient of the cost function does fall inversely with the dimension of the effective DLA for 2-designs of the dynamical Lie group. As we will show, the Heisenberg picture and the symmetries of the circuit's action on the observable are more suitable for explaining this phenomenon. This will lead to intuitive and commonplace conditions on the observable that are sufficient for this connection to hold. To show the validity of the 2-design assumption in practice, we show that fast mixing occurs for DLAs with polynomial dimensions, and we experimentally verify our formulae for the quantum compound ansatz.

## Results

### General framework

VQAs consist of optimizing the parameters of parameterized circuits of the form given in the below definition:

**Definition 2.1.** (Periodic ansatz) A periodic ansatz constructed from Hermitian generators $\{\tilde{\mathbf{H}}_1, \ldots, \tilde{\mathbf{H}}_K\}$ consists of a unitary of the form

$$\mathbf{U}(\boldsymbol{\theta}) = \prod_{l=1}^{L} \prod_{k=1}^{K} e^{-\theta_{(l,k)} i \tilde{\mathbf{H}}_k}, \tag{1}$$

an initial state $\rho = \mathbf{U}_0 |\mathbf{0}\rangle\langle\mathbf{0}| \mathbf{U}_0^\dagger$, and a Hermitian measurement operator $\mathbf{O}$.

The output of a VQA is the parameter-dependent expectation value $\langle\mathbf{O}\rangle_\rho = \text{Tr}\{\mathbf{U}(\boldsymbol{\theta})\rho\mathbf{U}^\dagger(\boldsymbol{\theta})\mathbf{O}\}$, known as the cost function.

For $n$-qubits, the set of $\mathbf{U}(\boldsymbol{\theta})$ lies in the unique connected subgroup of $SU(2^n)$, called the *dynamical Lie group*[32]. It is the subgroup associated with the real span of the Lie closure (i.e., closure under taking commutators) of the generators:

$$\mathfrak{g} := \text{span}_{\mathbb{R}}\langle i\tilde{\mathbf{H}}_1, \ldots, i\tilde{\mathbf{H}}_K\rangle_{\text{Lie}}, \tag{2}$$

which is known in the quantum control literature as the DLA[32]. We denote the dimension of $\mathfrak{g}$ as a real vector space by $d_\mathfrak{g}$.

We also informally define the notion of BP for quantum ansätze.

**Definition 2.2.** (Barren plateau) A class of quantum ansätze experiences a BP if the variance of the cost function gradient decays exponentially with system size, i.e., for all $(l, k)$,

$$\text{Var}_{\boldsymbol{\theta}\sim\nu}\left[\partial_{(l,k)}\langle\mathbf{O}\rangle_\rho\right] \in \mathcal{O}\left(\frac{1}{b^n}\right), \tag{3}$$

where the system size $n$ is the number of qubits and $b > 1$. Typically $\nu$ is the uniform distribution over the range of the parameters.

Note that in general a BP at initialization may not imply a BP throughout the training trajectory. However, in most cases when $\nu$ is the uniform distribution over parameters, the collection $\mathbf{U}(\boldsymbol{\theta})$ forms an approximate 2-design w.r.t. the Haar measure on the dynamical Lie group (this is made explicit in a later subsection), and due to Haar invariance, a BP at initialization implies a BP throughout training. A PQC that experiences a BP is also called untrainable, which follows from the gradient being computationally infeasible to estimate to arbitrary precision. Otherwise, if the variance only falls as $\Omega(1/\text{poly}(n))$, then the PQC is trainable.

## DLA - BP connection

It has been conjectured that the dimension of the DLA plays a crucial role in characterizing the trainability of VQAs. More specifically, the following conjecture linking trainability and DLA dimension was put forward:

**Conjecture 2.3.** (Conjecture 1 in ref. 22, paraphrased) The scaling of the variance of the partial derivatives of the cost function is inversely proportional to the dimension of the DLA:

$$\text{Var}_{\boldsymbol{\theta}\sim\nu}\left[\partial_{(l,k)}\langle\mathbf{O}\rangle_\rho\right] \in \mathcal{O}\left(\frac{1}{\text{poly}(d_\mathfrak{g})}\right). \tag{4}$$

In this work, we provide a proof of this conjecture. We emphasize that our results show a more explicit scaling of the variance with the DLA dimension, instead of just an upper bound. Thus, our results shed light on when stronger versions of the above conjecture hold, e.g., $\Theta(\frac{1}{\text{poly}(d_\mathfrak{g})})$. However, this depends on the initial state and observable, since the DLA dimension may not always be the quantity dominating the decay.

It turns out that the connection holds for a certain class of ansätze, which we term the class of *Lie algebra supported ansatz* (LASA).

**Definition 2.4.** (Lie algebra supported ansatz) A Lie algebra supported ansatz (LASA) is a periodic ansatz where the measurement operator $\mathbf{O}$ is such that $i\mathbf{O}$ belongs to the dynamical Lie algebra associated with the circuit generators $\{i\tilde{\mathbf{H}}_1, \ldots, i\tilde{\mathbf{H}}_K\}$.

In Fig. 1, we display our main result, which shows that the variance of the gradient has a direct dependence on DLA dimension for LASAs. As will be made rigorous later, by construction, the action of a LASA on its observable will decompose into invariant subspaces (corresponding to preserved symmetries) each of dimension at most $d_\mathfrak{g}$.

While we introduce restrictions on the observable, we note that our results are still far-reaching. This is because LASAs include many commonly used PQCs such as the Hamiltonian variational ansatz (HVA)[33] and quantum alternating operator ansatz (QAOA)[2,34]. We also note that all LASAs are equivariant quantum neural networks (EQNNs)[35]. However, an EQNN is not necessarily a LASA, since there are equivariant operators that may not lie in the DLA. This is because equivariance is defined with respect to a symmetry group of the quantum data, and one could imagine a situation in which the circuit has a small DLA such that other equivariant operators exist outside the DLA.

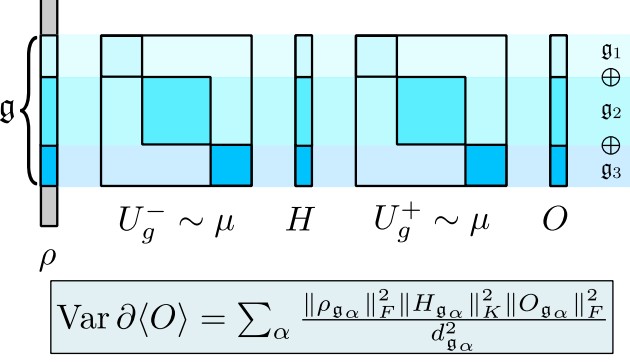

**Fig. 1 | Illustration of the main result.** For the gradient variance, when the observable is in the dynamical Lie algebra (DLA), as in the case of Lie algebra supported ansatz, only the components of $\rho$ in the DLA matter, and everything can be computed in the adjoint representation. Specifically, the subscript $\alpha$ for operators corresponds to their orthogonal projection onto the simple ideal $\mathfrak{g}_\alpha$. When the DLA has multiple ideals, each ideal individually contributes a term to the variance.

## Representation theoretic notation

The following presents the notation used throughout the paper and assumes familiarity with Lie groups and representation theory. The unfamiliar reader is directed to the Supplementary Information, where we briefly introduce Lie groups and representation theory.

Our focus will be a compact, connected Lie group $G$. The corresponding compact Lie algebra will be denoted by $\mathfrak{g}$. The notation $V$ will represent an arbitrary finite-dimensional inner product space over $\mathbb{C}$ or $\mathbb{R}$. If a result does not specify which field is used, then either can be assumed. Additionally, $\mathcal{U}(V)$ will denote the group of isometries on $V$ (i.e., depending on the field, either the unitary group or orthogonal group), and $\mathfrak{u}(V)$ will denote the set of skew-Hermitian operators on $V$. For either $\mathbb{R}$ or $\mathbb{C}$, we will use $\phi : G \to \mathcal{U}(V)$ to denote a unitary representation of the group $G$ and $d\phi : \mathfrak{g} \to \mathfrak{u}(V)$ to denote the differential or Lie algebra representation. We will frequently use the notation $\mathbf{U}_g$ to denote the element $\phi(g) \in \mathcal{U}(V)$ for some $g \in G$ when the representation $\phi$ and space $V$ are clear from the context.

Recall that the adjoint representation of a Lie group $G$ is the homomorphism:

$$\forall g \in G, \mathrm{Ad}_g(k) := gkg^{-1} \in \mathfrak{g}, \forall k \in \mathfrak{g}, \tag{5}$$

and the adjoint representaton of a Lie algebra is the homomorphism:

$$\forall h \in \mathfrak{g}, \mathrm{ad}_h(k) := [h,k] \in \mathfrak{g}, \forall k \in \mathfrak{g}. \tag{6}$$

For compact simple Lie algebras, since all trace forms are related by a real factor, we define a scaling constant $I_\phi$ that we call the *index of the representation* (w.r.t. the standard representation) such that:

$$-\mathrm{Tr}(d\phi(e_i)d\phi(e_j)) = I_\phi \delta_{ij}, \tag{7}$$

for $\{e_i\}$ a basis for $\mathfrak{g}$ satisfying:

$$-\mathrm{Tr}(e_i e_j) = \delta_{ij}. \tag{8}$$

The constant $I_\phi$ is the same as (twice) the Dynkin index for irreducible representations[36].

For compact simple Lie algebras we consider a few norms induced by the trace forms. For any $a \in \mathfrak{g}$, we define the *standard norm* to be

$$\|a\|_\mathfrak{g}^2 = -\mathrm{Tr}(a^2), \tag{9}$$

the *Killing norm* $\|a\|_K^2$ to be the norm induced by the Killing form (trace form associated with the adjoint representation), and more generally, for an arbitrary Lie algebra representation $d\phi$, we denote the usual Frobenius norm by $\|d\phi(a)\|_F^2$. All are related in the natural way via the associated index of the representation, as defined earlier. Specifically, for arbitrary $d\phi$:

$$\|d\phi(a)\|_F = I_\phi \|a\|_\mathfrak{g}^2 \tag{10}$$

$$\|d\phi(a)\|_K^2 = \|a\|_K^2 = I_{\mathrm{Ad}} \|a\|_\mathfrak{g}^2 = \frac{I_{\mathrm{Ad}}}{I_\phi} \|d\phi(a)\|_F^2. \tag{11}$$

For an arbitrary $\mathbf{X} \in \mathfrak{u}(V)$, we define $\mathbf{X}_\mathfrak{g}$ to be the orthogonal projection under the Frobenius inner product onto $d\phi(\mathfrak{g})$.

Lastly, throughout the paper, all integration, e.g. $\int_G f(g)dg$, is with respect to the Haar measure $\mu$ for $G$. The notation $\mu^{\otimes 2}$ will denote the product Haar measure.

Let us now place these notions in the context of VQAs. The vector space $V$ on which the group acts is the $n$-qubit Hilbert space $\mathbb{C}^{2^n}$. In general the PQC's dynamical Lie group will be $\phi(G)$ with $\phi$ a faithful (injective) representation and this is what we will assume here.

In practice however one always can take $\phi$ to be the identity map, identifying $G$ with the dynamical group and $\mathfrak{g}$ with the DLA, without invalidating the results.

In this abstract setting there is no notion of parameter space and hence the PQC gradient $\partial_{(l,k)}\langle \mathbf{O} \rangle_\rho$ is not well defined. Thus, we introduce the following parameter-independent quantity associated with any compact, connected Lie group:

**Definition 2.5.** (Abstracted gradient) Let $G$ be a compact, connected Lie group with representation $\phi : G \to \mathcal{U}(V)$. In addition, let $h \in \mathfrak{g}$ and $i\mathbf{O}, i\mathbf{A} \in \mathfrak{u}(V)$. We define the *abstracted gradient* to be the following quantity:

$$\partial \langle \mathbf{O} \rangle_\mathbf{A} := \mathrm{Tr}\{\mathbf{U}_{g^-}^\dagger \mathbf{A}\mathbf{U}_{g^-}[\mathbf{H}, \mathbf{U}_{g^+}\mathbf{O}\mathbf{U}_{g^+}^\dagger]\}, \tag{12}$$

where $\mathbf{U}_{g^\pm} := \phi(g^\pm)$ for arbitrary $g^+, g^- \in G$, and $\mathbf{H} = d\phi(h)$.

Note that now we set the generators to be skew-Hermitian. The connection between abstracted and PQC gradients is clear for the periodic ansatz in Definition 2.1: for any parameter $\theta_{(l,k)}$ the PQC gradient will be equivalent to an abstracted gradient, with $\mathbf{U}_{g^-}$ ($\mathbf{U}_{g^+}$) being the unitaries preceding (following) the unitary $e^{-\theta_{(l,k)}\mathbf{H}_k}$ in the circuit.

In our calculations we will look at second moments of the abstracted gradient for $(g^+, g^-) \sim \mu^{\otimes 2}$. This will accurately model the experimental behavior if for any $\theta_{(l,k)}$ the ansatz takes the form $\mathbf{W}^{(L)}e^{-\theta_{(l,k)}\mathbf{H}_k}\mathbf{W}^{(R)}$ with $\mathbf{W}^{(L/R)}$ random unitaries forming independent 2-designs for $\phi(G)$.

For a sufficiently deep periodic ansatz, the assumption is valid for parameters in the middle of the PQC whenever randomly initialized, polynomially-sized periodic ansätze form approximate 2-designs.

It has been shown that this holds for $\mathfrak{g} = \mathfrak{su}(2^n)$ or $\mathfrak{so}(2^n)$ and when all generators are in the Pauli group[37]. It has been widely assumed in literature that this result still holds for ansatz with different DLAs, with only numerical evidence. The following result answers this in the affirmative for LASA with polynomially-sized DLA, showing that rapid mixing to 2-design still holds when we sample generators from a basis for the DLA.

**Theorem 2.6.** (Rapid mixing for polynomial DLA) Consider an orthogonal basis of skew-Hermitian generators $\mathcal{A} := \{\mathbf{B}_1, \ldots, \mathbf{B}_{d_\mathfrak{g}}\}$ for the DLA with the property that the unitary $e^{-\theta\mathbf{B}_k}$ corresponding to a generator $\mathbf{B}_k$ is $t_k$-periodic. In addition, suppose that $d_\mathfrak{g} = \mathcal{O}(\mathrm{poly}(n))$. Consider a LASA formed by applying evolutions $e^{-\theta_k\mathbf{B}_k}$ where $\mathbf{B}_k$ is selected uniformly at random from the set $\mathcal{A}$ and the parameter $\theta_k$ uniformly from $[0, t_k]$. Then, the ansatz is an $\epsilon$-approximate 2-design for the dynamical group $\mathcal{G}$ after $\mathcal{O}(\mathrm{poly}(n)\log(1/\epsilon))$ layers.

Note that this result only focuses on bounding the spectral gap of the walk, i.e., a "layer" is a single application of evolution, and does not include the cost of implementing the evolutions $e^{-\theta\mathbf{B}_k}$ in terms of basis gates. The proof of the above result and its generalization to $t$-designs for arbitrary LASA are in the Supplementary Information and are based on techniques used by ref. 37 and earlier works. Such random walks have been known to converge for some time[38], and convergence to Haar for exponential DLA is not efficient. However, the above result makes the spectral gap dependence explicit. As a comparison, $\mathcal{O}(\log(1/\epsilon))$ layers suffice for random Pauli rotations to approximate a 2-design for SU($2^n$), as shown by ref. 37. The approach of studying BPs with 2-designs is standard, e.g. see ref. 18. Furthermore, as we have shown, it is theoretically motivated in the case of independent, uniformly distributed parameters. However, there may still be settings where the 2-design assumption fails and where our results will not hold, for example, other initialization schemes or correlated parameters. Interestingly, there is evidence that both may avoid BPs[39,40], however they do not investigate this research direction further.

Inspired by our overall goal of analyzing BPs in PQCs, we seek to compute the quantity

$$\mathrm{GradVar} := \mathrm{Var}_{(g^+,g^-)\sim\mu^{\otimes 2}}\left[\partial\langle\mathbf{O}\rangle_\rho\right]$$
$$= \mathbb{E}_{(g^+,g^-)\sim\mu^{\otimes 2}}\left[\left(\partial\langle\mathbf{O}\rangle_\rho\right)^2\right] - \left(\mathbb{E}_{(g^+,g^-)\sim\mu^{\otimes 2}}\left[\partial\langle\mathbf{O}\rangle_\rho\right]\right)^2, \quad (13)$$

where $\mu$ is the unique Haar measure over $G$ and $\rho$ is the initial quantum state to which all elements of the dynamical group are applied. $\mathbb{E}_{(g^+,g^-)\sim\mu^{\otimes 2}}[\partial\langle\mathbf{O}\rangle_\rho]$ can be shown to be zero is general (see the Supplementary Information), and thus in practice, we focus on the second moment:

$$\mathrm{GradVar} = \mathbb{E}_{(g^+,g^-)\sim\mu^{\otimes 2}}[(\partial\langle\mathbf{O}\rangle_\rho)^2]. \quad (14)$$

Using Definition 2.2, a BP occurs when the following holds:

$$\mathrm{GradVar} \in \mathcal{O}\left(\frac{1}{b^n}\right), \quad b>1. \quad (15)$$

This is the phenomenon that our methods will seek to probe for the specific case of LASAs.

Lastly, we formally define what we mean by symmetries in the Schrödinger and Heisenberg evolution pictures. Schrödinger symmetries refer to invariant linear subspaces $V_s$ of states that are preserved by evolutions generated by the dynamical Lie group $\mathcal{G}$, i.e., $\forall\mathbf{U}\in\mathcal{G}, \mathbf{U}V_s \subseteq V_s$. Heisenberg symmetries refer to invariant linear subspaces of observables $V_h$ preserved by evolutions generated by having the dynamical Lie group $\mathcal{G}$ act via conjugation, i.e., $\forall\mathbf{U}\in\mathcal{G}, \mathbf{U}V_h\mathbf{U}^\dagger \subseteq V_h$. If the observables lie in the DLA, i.e. the LASA case, then this is the adjoint representation.

## Theory of BPs for LASA

We now present our theoretical contributions, which connect the Lie algebra dimension to the scaling of the gradient variance. We note that norms involving the Hermitian observable $\mathbf{O}$ and the skew-Hermitian generator $\mathbf{H}$ have a few interpretations as mentioned in the previous subsection. However, to be concise and for readability, we present the results in only one form.

We start by recalling that all compact Lie algebras (and thus groups) are reductive.

**Definition 2.7.** (Reductive Lie algebra[41]) A Lie algebra $\mathfrak{g}$ is reductive if the adjoint representation is completely reducible, i.e., $\mathfrak{g}$ has the following decomposition as a direct sum of Lie algebras:

$$\mathfrak{g} = \bigoplus_\alpha \mathfrak{g}_\alpha \oplus \mathfrak{c}, \quad (16)$$

where each $\mathfrak{g}_\alpha \subset \mathfrak{g}$ is a simple ideal and $\mathfrak{c} \subset \mathfrak{g}$ is the center of $\mathfrak{g}$. Note that if $G$ is simply connected then $\mathfrak{c} = \{0\}$.

This property is essential for proving our main result, as it allows us to extend our expression (Theorem 2.8) for the gradient variance for simple Lie groups to the general compact case. If $\mathfrak{g}$ is compact, then the $\mathfrak{g}_\alpha$ will be compact as well[42].

Note that this notion of reducibility is related to what has appeared in prior works, e.g., refs. 22,23,30, the differences are mainly as to whether the group acts on the observable or state. We discuss this in detail in the last subsection.

Next, we present our expression for the variance of the gradient for compact simple groups that applies to each $\mathfrak{g}_\alpha$ in Equation (16).

**Theorem 2.8.** (Simple group variance) Let $G$ be a compact, connected simple Lie group with Lie algebra $\mathfrak{g}$. Suppose $\phi$ is a finite-dimensional unitary representation of $G$. In addition, $o,h \in \mathfrak{g}$, $i\mathbf{O} = d\phi(o)$, $\mathbf{H} = d\phi(h)$

and $\rho$ a density matrix. Then the following holds:

$$\mathrm{GradVar} = \frac{\|\mathbf{H}\|_{\mathrm{K}}^2 \|\mathbf{O}\|_{\mathrm{F}}^2 \|\rho\|_{\mathrm{F}}^2}{d_\mathfrak{g}^2}. \quad (17)$$

If $G$ is compact, one can use the fact that it is reductive and apply Theorem 2.8 to each of the compact simple ideals to obtain the following:

**Theorem 2.9.** (Compact group variance) Let $G$ be a compact and connected Lie group with Lie algebra $\mathfrak{g}$. Suppose $\phi$ is a finite-dimensional unitary representation, $o,h \in \mathfrak{g}$, $i\mathbf{O} = d\phi(o)$, $\mathbf{H} = d\phi(h)$, and $\rho$ is a density matrix. Then the following holds:

$$\mathrm{GradVar} = \sum_\alpha \frac{\|\mathbf{H}_{\mathfrak{g}_\alpha}\|_{\mathrm{K}}^2 \|\mathbf{O}_{\mathfrak{g}_\alpha}\|_{\mathrm{F}}^2 \|\rho_{\mathfrak{g}_\alpha}\|_{\mathrm{F}}^2}{d_{\mathfrak{g}_\alpha}^2}. \quad (18)$$

Note that the center $\mathfrak{c}$ does not contribute to the variance.

As mentioned in the Introduction, the above theorem is the central result of the paper. It shows that under the assumption of a LASA we can get a precise mathematical expression for the gradient variance. Notably, this expression is in terms of quantities that are intimately linked with the Lie algebra and the representation and are well characterized for all simple algebras.

## Interpretation of results

The three norms in the numerator of Equation (18) can be viewed as effectively measuring the support that each operator has on the simple ideal $d\phi(\mathfrak{g}_\alpha)$. Specifically, $\|\mathbf{O}_{\mathfrak{g}_\alpha}\|_{\mathrm{F}}$ and $\|\rho_{\mathfrak{g}_\alpha}\|_{\mathrm{F}}$ being Frobenius norms can be interpreted as generalized measures of purity with respect to $d\phi(\mathfrak{g}_\alpha)$. This concept was actually first introduced in ref. 43. A similar interpretation is also valid for the Killing norm $\|\mathbf{H}_{\mathfrak{g}_\alpha}\|_{\mathrm{K}}$, however, this time the relevant representation of $\mathfrak{g}_\alpha$ is the adjoint representation, and so the norm is scaled by the ratio of the indices as in Equation (11).

If one is still uncomfortable with the Killing norm, we note that $\|\mathbf{H}_{\mathfrak{g}_\alpha}\|_{\mathrm{K}}^2 \leq 2d_{\mathfrak{g}_\alpha}\|\mathbf{H}_{\mathfrak{g}_\alpha}\|_{\mathrm{F}}^2$ (see the Supplementary Information), and so one gets the following upper bound:

$$\mathrm{GradVar} \in \mathcal{O}\left(\sum_\alpha \frac{\|\mathbf{H}_{\mathfrak{g}_\alpha}\|_{\mathrm{F}}^2 \|\mathbf{O}_{\mathfrak{g}_\alpha}\|_{\mathrm{F}}^2 \|\rho_{\mathfrak{g}_\alpha}\|_{\mathrm{F}}^2}{d_{\mathfrak{g}_\alpha}}\right), \quad (19)$$

which presents the result in terms of more familiar quantities, i.e., Frobenius norms. In addition, we now see that Conjecture 2.3 is explicitly proven (and indeed significantly generalized) for LASA.

From Equation (18) or (19), we infer that a BP can only occur whenever at least one of the terms in the expression leads to exponential decay. More specifically, the gradients will decay exponentially under any of these conditions: the state has exponentially small support over the Lie algebra; the state, the measurement operator, and the generator are mostly supported on a subalgebra, $\mathfrak{g}_\alpha$, the dimension of which is exponentially large; or the support of the state, measurement operator and generator are mutually incompatible on the subalgebras, in the sense that all terms vanish. The second condition amounts to the conjecture of ref. 22, while the last is a novel prediction of this work, which only occurs in the strict semisimple case.

Lastly, we conclude with some details on how one might use our results in practice to forecast gradient variance scaling without access to a quantum computer. The main goal is to find a basis for the DLA and compute its structure constants in $\mathcal{O}(\mathrm{poly}(d_\mathfrak{g}))$ time. Since the generators and observables will typically be linear combinations of Pauli strings, one can utilize symbolic computation to reason about the decomposition of $\mathfrak{g}$ into simple ideals. A basis for the DLA can be obtained by computing nested commutators symbolically and checking for linear independence as done in ref. 22. In summary, as input we are given the Hermitian generators used in the ansatz.

We proceed by computing pairwise commutators, until we find no new linearly independent elements. If our current estimate for the basis has $k$ elements, then we need to compute $\binom{k}{2}$ pairwise commutators, and we need at most $d_\mathfrak{g}$ iterations. This leads to $\mathcal{O}(d_\mathfrak{g}^3)$ pairwise commutators in total.

Next, using the basis $\{\mathbf{E}_k\}_{k=1}^{d_\mathfrak{g}}$ for the DLA obtained from the process just discussed expressed as sums of Pauli strings, we compute the $d_\mathfrak{g} \times d_\mathfrak{g}$ matrices for each operator $\mathrm{ad}_{i\mathbf{E}_k}$ in the basis $\{\mathbf{E}_k\}_{k=1}^{d_\mathfrak{g}}$. We denote these matrices by $\widehat{\mathrm{ad}_{i\mathbf{E}_k}}$, which contain the structure constants. The next step is to simultaneously block diagonalize the $\widehat{\mathrm{ad}_{i\mathbf{E}_k}}$, which will reveal bases for the simple ideals. This can be done in $\mathcal{O}(\log(d_\mathfrak{g})\mathrm{poly}(d_\mathfrak{g}))$ by diagonalizing $\widehat{\mathrm{ad}_{i\mathbf{E}_1}}$, and then finding invariant subspaces preserved by $\widehat{\mathrm{ad}_{i\mathbf{E}_2}}$. Then repeat this procedure for each smaller block that was found for $\widehat{\mathrm{ad}_{i\mathbf{E}_2}}$ in the previous step, and so on. We can compute the $\|\mathbf{O}_{\mathfrak{g}_\alpha}\|_F^2$ and $\|\mathbf{H}_{\mathfrak{g}_\alpha}\|_K^2$ norms symbolically. If $\{\mathbf{A}_k\}_{k=1}^{d_{\mathfrak{g}_\alpha}}$ is a basis for the ideal $\mathfrak{g}_\alpha$, which can be expressed in terms of sums of Pauli strings given our assumption, then the norm $\|\rho_{\mathfrak{g}_\alpha}\|_F^2 = \sum_{k=1}^{d_{\mathfrak{g}_\alpha}} \mathrm{Tr}(\mathbf{A}_k^{\otimes 2} \rho^{\otimes 2})$ can be computed classically for product input states.

However, we cannot yet claim that the overall computational complexity is polynomial in $d_\mathfrak{g}$, as we typically express operators in the Pauli basis, and computing pairwise commutators can cause the support on the Pauli basis to grow, in the worst case, exponentially with the number of iterations. Simply put, there may be no way to express the basis elements compactly. This is the same challenge with the classical simulation technique $\mathfrak{g}$-sim[27] and is currently unclear whether it can be overcome in general. Assuming that the growth of support of nested commutators in the Pauli basis does grow polynomially with the number of iterations, then we have a procedure with a runtime that is a (potentially large) polynomial in $d_\mathfrak{g}$. If the DLA dimension is polynomial, then it is an overall polynomial-time process. This can at least be done at small scales to probe the scaling of the gradient variance.

The analysis so far assumed no a priori knowledge about the DLA. The situation radically improves when the DLA isomorphism class is known. Then exact variance calculation with our formula can become a relatively straightforward task, as we shall see in the next subsection.

**Variance computation for quantum compound Ansatz**

The quantum compound ansatz is a quantum representation on $n$ qubits ($2^n$-dimensional) of the Lie group SO($n$) or SU($n$)[30,44]. Given a general $g \in \mathrm{SU}(n)$ (SO($n$)), one can decompose it into a product of SU(2) (SO(2)) rotations on 2-dimensional subspaces, which are (generalized) Givens rotations:

$$\mathbf{U}_g = \prod_{(i,j) \in E} \mathbf{U}_{ij}^{\mathrm{Givens}}(g), \tag{20}$$

and are implemented using the fermionic beam splitter (FBS) gate defined in ref. 30.

The graph $E$ can have various topologies, for example a pyramid or a staircase. The circuit preserves Hamming weight, and the representation splits into subspaces corresponding to the different Hamming weights. The analysis of the gradient variance for a more general class of Hamming weight-preserving unitaries appears in ref. 45.

One can check that the appropriate representation for the generators of a SU(2) Givens rotation between qubit $i$ and $j$

$$h_x^{ij} = -\frac{i}{4}(\sigma_x^i \otimes \sigma_x^j + \sigma_y^i \otimes \sigma_y^j) \otimes \sigma_z^{\otimes|i-j-1|} \tag{21}$$

$$= d\phi\left(-\frac{i}{2}X^{(ij)}\right) \tag{22}$$

$$h_y^{ij} = -\frac{i}{4}(\sigma_y^i \otimes \sigma_x^j - \sigma_x^i \otimes \sigma_y^j) \otimes \sigma_z^{\otimes|i-j-1|} \tag{23}$$

$$= d\phi\left(-\frac{i}{2}Y^{(ij)}\right) \tag{24}$$

$$h_z^{ij} = -\frac{i}{4}(\sigma_z^i - \sigma_z^j) = d\phi\left(-\frac{i}{2}Z^{(ij)}\right), \tag{25}$$

where $X^{(ij)}, Y^{(ij)}, Z^{(ij)}$ act as the Pauli operators $\sigma_x, \sigma_y, \sigma_z$ on the $2 \times 2$ block formed by $i$ and $j$, respectively, and are zero otherwise. They are elements of $\mathfrak{su}(n)$, and $\phi$ is the direct sum of the alternating representations for $k = 1, \ldots, n$, i.e.:

$$V = \bigoplus_{k=1}^n \wedge^k \mathbb{C}^n. \tag{26}$$

Note that the norm of each of these generators in $\mathfrak{g}$ is $1/2$. Importantly, while the set of generators spans the representation of $\mathfrak{g}$, since it is larger than the dimension of $\mathfrak{g}$ it is a not linearly independent set. Note the extra $\sigma_z$'s in the definition of $h_x$ and $h_y$ are reminiscent of the string of $\sigma_z$ in the Jordan–Wigner encoding, only that here they are needed for the algebra to close. The SO case is generated by the $h_y^{ij}$ elements only.

To clarify why the ansatz is subspace uncontrollable, we can consider the Hamming weight $n/2$ subspace. On this subspace, the DLA is isomorphic to $\mathfrak{su}(n)$, while the Lie algebra of the full space of unitary operators on this subspace is isomorphic to $\mathfrak{su}\left(\binom{n}{n/2}\right)$, hence the compound ansatz cannot enact all unitary transformations.

Before proceeding we present a mixing time result to $t$-design for the quantum compound ansatz that is tighter than Theorem 2.6.

**Theorem 2.10.** (Rapid mixing for Compound Ansatz) Consider an $n$-qubit quantum compound ansatz that is a LASA constructed using the set of generators $\{X^{(ij)}, Y^{(ij)}, \sum_{i=1}^j Z^{(ij)}\}$ with rotations angles chosen uniformly at random. Then, for $t \le n/2$, the ansatz is an $\epsilon$-approximate $t$-design for the dynamical group SU($n$) after $\mathcal{O}(tn\log(1/\epsilon))$ layers.

Of course, for BPs $t = 2$ is the main interest. The proof follows simply from a generalization of Theorem 2.6 and is left to the Supplementary Information. Note that for the chosen set of generators some of the randomly chosen angles are not independent (i.e., the $\sum_{i=1}^j Z^{(ij)}$ type generators).

The following three results utilize our theory of BPs for LASA to show that the quantum compound ansatz can be BP-free under uniform initialization.

**Theorem 2.11.** For a quantum compound ansatz that is also LASA, if the initial state is a computational basis state, then the following holds:

$$\mathrm{GradVar} \in \Omega\left(\frac{1}{n^3}\right). \tag{27}$$

The conclusion is that SU compound layers with Lie algebra-supported measurements do not have BPs for any fixed Hamming weight computational basis state. Note that computational basis states of the same Hamming weight are in an irreducible subspace of the tensor product representation (see the Supplementary Information).

Next, we consider the uniform superposition state $|\psi\rangle = |+\rangle^{\otimes n}$ and show that the quantum compound ansatz is still BP-free. In addition, in this case, the variance decays exactly with the DLA dimension $n^2 - 1$.

**Theorem 2.12.** For a quantum compound ansatz that is also LASA, if the initial state is a uniform superposition of all computational basis

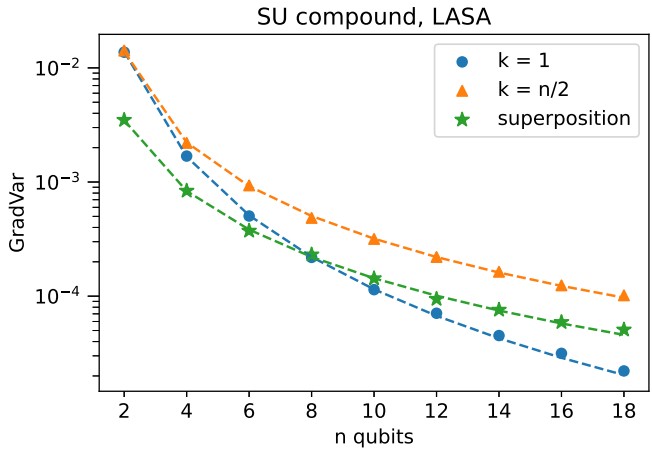

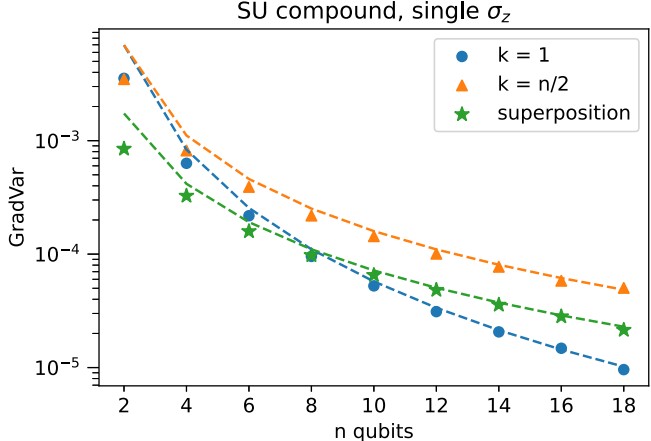

**Fig. 2 | Gradient variance scaling for SU compound layers, observable in Lie algebra.** Dots are numerical results while dotted lines are analytical predictions using the equations in the text. Showing results for computational basis input states of Hamming weight 1 and $n/2$ and the uniform superposition state $|+\rangle^{\otimes n}$, for $n$ number of qubits ranging from 2 to 18 in steps of 2. The measurement operator is $-ih_{12}^z = (\sigma_1^z - \sigma_2^z)/4$. Accounting for the randomness of initialization, there is good agreement of numerical results with the predictions. The error bars are too small to plot. Additional information on the numerics is in the Supplementary Information.

**Fig. 3 | Gradient variance scaling for SU compound layers, observable not in Lie algebra.** The setup is identical to the Lie algebra supported ansatz (LASA) case, except that here the measurement operator is a single $\sigma_z/4$. We show the analytical prediction derived from the LASA case as explained in the text, and therefore we see a disagreement with numerics, implying that the covariance term is nonzero. Still, the scaling is similar, and additionally, the numerics converge to the prediction at larger system sizes. The error bars are too small to plot. Additional information on the numerics is in the Supplementary Information.

states, then the following holds:

$$\text{GradVar} \in \Theta\left(\frac{1}{n^2}\right). \tag{28}$$

Thus, we also have no BP with the initial state being the uniform superposition. We numerically verified the predictions for the various initial states as shown in Fig. 2.

Finally, we see how the result can be extended to cover single-qubit measurements.

**Corollary 2.12.1.** For a quantum compound ansatz with an observable that is composed of single-qubit measurements, and if the initial state is a computational basis state or the uniform superposition of all computational basis states, then the following holds:

$$\text{GradVar} \in \Omega\left(\frac{1}{\text{poly}(n)}\right). \tag{29}$$

We verify these predictions in Fig. 3.

This answers an open question proposed in ref. 31. As a final note, even though $\sigma_i^z$ does not lie in the DLA, single-qubit expectations of observables with respect to the compound ansatz starting from a product state are still known to be classically simulatable[46].

Lastly, we present another setting in which the observable does not lie in the DLA, but this time, the quantum compound ansatz has a BP.

**Theorem 2.13.** For the quantum compound ansatz if the initial state is a computational basis state with Hamming weight $\frac{n}{2}$ and the observable is a rank-one projector onto another computational basis state in this space, then

$$\text{GradVar} \in \mathcal{O}\left(\binom{n}{n/2}^{-1}\right). \tag{30}$$

We verify the scaling in Fig. 4.

Intuitively, the above decay comes from the fact our choice of observable and initial state are rank-one projectors, and thus, the overlap of traceless parts of both operators will spread across an

exponentially large subset of $\mathfrak{su}$. Theorem 2.13 is interesting because the compound ansatz is not very expressive and the depth of the circuit exceeds the shallow regime of $\mathcal{O}(\log(n))$[47]. We note that the cost function we choose is still global.

The details of how the numerical results were obtained are described in the Supplementary Information.

**Comparison with previous approaches**

As mentioned in the Introduction, previous approaches have taken a state-first or Schrödinger picture viewpoint. Specifically, under the action of $G$, the quantum state space $V$ will decompose into invariant subspaces:

$$V = \bigoplus_\kappa V_\kappa, \tag{31}$$

each of which is acted upon by the subrepresentation $\phi_\kappa(G)$. This decomposition is in line with the symmetries that the ansatz obeys, i.e., its commutant[35]. If the initial state $\rho \in V_\kappa$, then since $G$ preserves this space, the variance calculation is restricted to integrating over $\phi_\kappa(G)$. If the restriction of the DLA $d\phi_\kappa(\mathfrak{g})$ to the invariant subspace is isomorphic to $\mathfrak{su}(\dim V_\kappa)$, then one says that PQC is *subspace controllable* on $V_\kappa$, otherwise, it is *subspace uncontrollable*. The calculation is possible in the subspace-controllable setting via the Schur-Weyl duality[22], but the subspace uncontrollable setting poses significant obstacles to the calculation of the second moment (Equation (14)) using this approach.

In our setting we are instead using the Heisenberg picture and, assuming LASA, considering the action of $d\phi(\mathfrak{g})$ on itself via conjugation, so $V = d\phi(\mathfrak{g})$ in this case and $d\phi$ is the adjoint representation. Notice that if the DLA is reductive (Equation (16)) and $\phi$ is faithful (injective), the decomposition of $V$ respects the decomposition into simple ideals:

$$d\phi(\mathfrak{g}) = d\phi\left(\bigoplus_\kappa \mathfrak{g}_\kappa\right) = \bigoplus_\kappa d\phi(\mathfrak{g}_\kappa) = \bigoplus_\kappa d\phi(\mathfrak{g})_\kappa. \tag{32}$$

Thus, the Lie algebra being reductive implies that the adjoint representation splits into irreducible invariant subspaces, which are precisely the simple ideals $d\phi(\mathfrak{g}_\kappa)$. As detailed in Methods, this is sufficient to calculate the second moment for any compact Lie group.

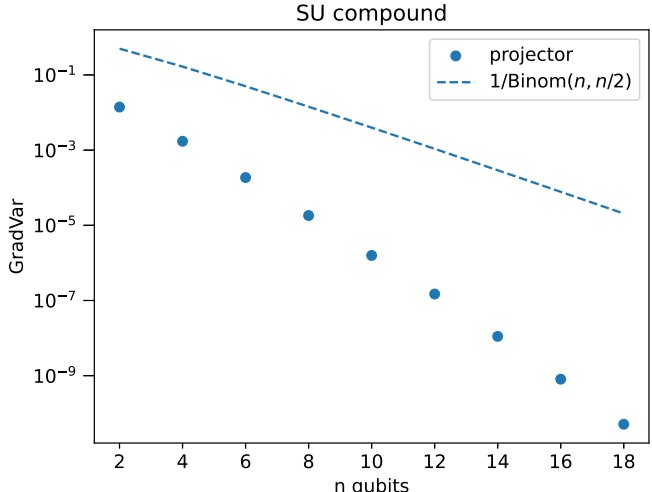

**Fig. 4 | Gradient variance scaling for SU compound layers, projective measurement.** Numerics for gradient variance of SU compound layer, with input state is a computation basis state of Hamming weight $n/2$ and the observable is a projection onto the same state. The resulting PQC is not a Lie algebra-supported ansatz, and indeed displays a barren plateau. We also show the upper bound of $\binom{n}{n/2}^{-1}$ which appears to be very loose. The error bars are too small to plot. Additional information on the numerics is in the Supplementary Information.

So in this setting, we always know the invariant subspaces and the representation acting on them, namely the $d\phi(\mathfrak{g}_\kappa)$ and the corresponding adjoint representation. This is a significant simplification from the Schrödinger picture approach and enables us to completely circumvent the obstacles posed by the subspace uncontrollable setting. Notice finally that now the invariant subspaces $d\phi(\mathfrak{g}_\kappa)$ reflect the symmetries that are preserved by the evolution of observable instead of the state, so while related this is a different concept of PQC symmetry than the one prior work had explored.

Lastly, we would like to emphasize that DLA does not always split into a direct sum over the decomposition of $V$ into $V_k$ for an arbitrary unitary representation. However, this does hold if $d\phi(\mathfrak{g})_\kappa$ for subspace $V_\kappa$ is simple, like it is for the adjoint and in ref. 23. More specifically, the condition implies that $d\phi(\mathfrak{g})_\kappa$ must then be $d\phi(\mathfrak{g}_\alpha)$ for some simple ideal $\mathfrak{g}_\alpha$.

## Discussion

In this work, we present a general framework for diagnosing the BP phenomenon in Lie algebra supported ansätze, which includes popular PQCs, such as HVA, QAOA, and various equivariant QNNs. Our main contribution is a method that explains the previously mysterious connection between the dimension of the DLA and the rate at which gradients decay. This method has enabled us to analyze the gradient variance for subspace uncontrollable circuits, such as the quantum compound ansätze, which was not previously possible with existing techniques from the literature.

We note that the kinds of circuits where the simulatability results of ref. 27 apply are exactly LASAs. In fact, many of the techniques employed here are similar. As the aforementioned paper links the dimension of the DLA to the performance of the classical simulation of expectations via their algorithm $\mathfrak{g}$-sim, we see that at least for LASAs there is a connection between the absence of vanishing gradients and simulatability, in the sense that a LASA with polynomial DLA can avoid BPs but is classically simulatable. Future work may look at the vanishing gradients in other symmetric settings like those of refs. 24–26, and at elucidating this connection more generally. We also note that our

results could be applied to the DLAs that have been classified by ref. 42.

Regarding general VQAs, when the observable has support outside of the DLA, we show in the Supplementary Information that the same techniques used in the LASA setting can be used to obtain the gradient variance expression for general ansatz. Unfortunately, it can be challenging to determine gradient variance scaling from these expressions in general. Characterizing the gradient variance in this setting would potentially allow for constructing ansätze that both do not have BPs and do not have classically simulated expectations. Existing literature has already shown that when the observable lies in the DLA and the DLA has polynomially growing dimension, then the computation of expectation values can be classically simulated. Potentially, the gradient variance can be shown to still scale inversely with the DLA dimension when the observable has only some small support outside of the DLA, as we have shown for the quantum compound ansatz (Corollary 2.12.1).

Lastly, BPs only correspond to one of two issues that plague VQAs. As mentioned earlier, like BPs, the convergence of VQAs has also only been theoretically characterized in the subspace-controllable setting[14]. Potentially, the framework we have developed can be applied to understanding the projected gradient dynamics that occur in the uncontrollable setting.

Note on ref. 48: During the writing of the manuscript, we became aware through a comment in ref. 42 that Michael Ragone et al. have independently obtained a proof of an extension of the conjecture in ref. 22. This was later released in ref. 48. We encourage the reader to review both papers for a richer picture of the solution, however we summarize here the most important differences between our works. The main one is that the work of Ragone et al. focuses on cost function concentration as opposed to concentration of the partial derivatives. The authors mention, by citing ref. 49, that loss function concentration implies concentration of the partial derivatives, and thus provide bounds. However, in our case, we obtain exact expressions for the variance of the partial derivatives, thus revealing the connection between the gradient variance scaling and the Killing norm of the generators. In addition, we include explicit formulae for the gradient variance for the quantum compound ansatz in commonly used settings, which leads to the novel prediction that it can avoid BPs under Haar initialization. Lastly, we include a discussion on the application of our techniques to observables that lie outside of the DLA. The work by Ragone et al., however, does include a broader discussion that links BPs in symmetric ansätze to other known causes of BPs, including cost function-induced[19] and noise-induced[20], and thus places the result into a wider context.

## Methods

In this section, we formally derive the connection between the DLA dimension and the gradient variance, leading to our theory of BPs. Specifically, we present the proofs of the majority of the theorems shown in the Results section, the rest are left to the Supplementary Information. The main tools that we utilize are the concepts of the adjoint representation and Schur orthogonality.

### The adjoint representation connection

We start by providing some explanation as to why the connection between the DLA dimension and BPs that agrees with existing numerical evidence is not obvious. It will be the adjoint representation that makes the relationship clear and allow for exact computation of the gradient variance that agrees with existing numerics.

As in earlier parts of the text, the dynamical group $\mathcal{G}$ associated to a periodic ansatz is a unitary representation of some other Lie group $G$. Thus, the representation $\phi: G \to \mathrm{SU}(2^n)$ corresponds to $G$ acting on the

$n$-qubit Hilbert space $V$ and $\phi(G) = \mathcal{G}$. Let $\mathcal{M}(\mathbb{C}, 2^{2n})$ denote the set of $2^{2n} \times 2^{2n}$ complex matrices.

Before proceeding we make a small note on the compactness of the dynamical group. While the dynamical group $\mathcal{G}$ is obviously connected, it may not be compact (due to lack of closure). An example is the irrational flow on a torus that occurs when the generators $\bar{\mathbf{H}}_i$ have at least two eigenvalues whose ratio is irrational. The action of these generators will lead to non-periodic orbits. Notice that such non-periodic ansätze can occur in principle, for example in QAOA on graphs with random weights. However, since any Lie subalgebra of $\mathfrak{su}(2^n)$ must be the direct sum of compact simple Lie subalgebras and its center[42,50], ignoring the center also leads to a compact, connected dynamical subgroup. Thus if $\mathcal{G}$ is not closed, this will be the compact dynamical subgroup we consider. Note that it is harmless to ignore the center, since the component of the observable in the center of $\mathfrak{g}$ does not evolve (in a Heisenberg sense) anyways.

The variance of the gradient, under Haar initialization, relies on the *second-moment operator*:

$$\mathcal{T} : \mathbf{A} \mapsto \int_G (\mathbf{U}_g \otimes \mathbf{U}_g) \mathbf{A} (\mathbf{U}_g^\dagger \otimes \mathbf{U}_g^\dagger) \, dg, \tag{33}$$

which orthogonally projects onto the set of commuting operators (i.e., commutant) of $\{\mathbf{U}_g \otimes \mathbf{U}_g : \forall g \in G\}$. Commutation implies that $\forall \mathbf{A} \in \mathcal{M}(\mathbb{C}, 2^{2n})$, $\mathcal{T}(\mathbf{A})$ must respect the decomposition of $V^{\otimes 2}$ into irreducible components (invariant subspaces). If $V^{\otimes 2}$ has the following decomposition into irreducible components (not grouping by multiplicity)

$$V^{\otimes 2} = \bigoplus_\lambda V_\lambda, \tag{34}$$

then

$$\int_G (\mathbf{U}_g \otimes \mathbf{U}_g) \mathbf{A} (\mathbf{U}_g^\dagger \otimes \mathbf{U}_g^\dagger) \, dg = \sum_\lambda \frac{\mathrm{Tr}[\mathbf{A} P_\lambda]}{\dim V_\lambda} P_\lambda, \tag{35}$$

for orthogonal projectors $P_\lambda$ onto $V_\lambda$. This projection can also be expressed in terms of the well-known Weingarten function[51,52]. Notice that the Lie algebra appears to play no role in this discussion. In addition, the inverse scaling with the dimension of each $V_\lambda$ is apparent. Furthermore, while a general theory of such integrals exists[53], they are quite challenging to tackle in practice. Most results in quantum information restrict to the case where $G = \mathrm{SU}(2^n)$, where the commutant is easy to characterize. Specifically, this leads to the well-known result that approximate 2-designs for $\mathrm{SU}(2^n)$ have BPs[18,22].

Fortunately, the integrals appearing in the theory of VQAs turn out to have substantial simplifications, which furnishes the connection to the dimension of the DLA in certain settings. Our results shed much-needed light on this apparently unintuitive phenomenon observed in practice. The first key insight is that $\mathbf{A}$ is always a tensor product of two operators, i.e., if $\mathbf{O}$ is the observable in the quantum circuit, then we get second-moment integrals with $\mathbf{A} = i\mathbf{O} \otimes i\mathbf{O}$, that is,

$$\int_G (\mathbf{U}_g i\mathbf{O} \mathbf{U}_g^\dagger) \otimes (\mathbf{U}_g i\mathbf{O} \mathbf{U}_g^\dagger) \, dg \tag{36}$$

$$= \int_G \mathrm{Ad}_g(i\mathbf{O}) \otimes \mathrm{Ad}_g(i\mathbf{O}) \, dg, \tag{37}$$

where the relation to the well-known adjoint representation, of $G$, i.e., $\mathrm{Ad}_g(i\mathbf{O}) = \mathbf{U}_g i\mathbf{O} \mathbf{U}_g^\dagger$, is apparent when $i\mathbf{O}$ lies in $\mathfrak{g}$. This simple observation is critical in enabling concise expressions for the variance of the gradient, revealing the inverse dependence on the dimension of the DLA. Specifically, given that the dimension of the adjoint representation is $d_\mathfrak{g}$ the reason for the scaling becomes more plausible.

Note that to connect back to (35), this can also be viewed as a projection of the subspace

$$S := \mathrm{span}_\mathbb{C}\{i\mathbf{O} \otimes i\mathbf{O} : i\mathbf{O} \in d\phi(\mathfrak{g})\} \subset \mathcal{M}(\mathbb{C}, 2^{2n}) \tag{38}$$

onto the commutant via an operator called the Casimir.

The (split quadratic) *Casimir operator*, $\mathbf{K}$, for representation $\phi$ is defined as:

$$\mathbf{K} = I_\phi^{-1} \sum_i \mathbf{E}_i \otimes \mathbf{E}_i, \tag{39}$$

where $\{e_i\}$ is an orthonormal basis under the standard norm for $\mathfrak{g}$ and $\mathbf{E}_i = d\phi(e_i)$. We can also use the Casimir to define an orthogonal projector, $P_\mathfrak{g}$, from the space of skew-Hermitian operators on $V$, i.e., $\mathfrak{u}(V)$, onto the subspace $d\phi(\mathfrak{g})$, which is useful when we are dealing with objects not completely supported on the Lie algebra:

$$\mathbf{X}_\mathfrak{g} := P_\mathfrak{g} \mathbf{X} = -\mathrm{Tr}_1((\mathbf{X} \otimes \mathbb{1})\mathbf{K}) \tag{40}$$

$$= -I_\phi^{-1} \sum_i \mathrm{Tr}(\mathbf{X} \mathbf{E}_i)\mathbf{E}_i, \tag{41}$$

$$\|\mathbf{X}_\mathfrak{g}\|_\mathrm{F}^2 = -\mathrm{Tr}((\mathbf{X} \otimes \mathbf{X})\mathbf{K}) = -I_\phi^{-1} \sum_i \mathrm{Tr}^2(\mathbf{X} \mathbf{E}_i) \tag{42}$$

where $\mathbf{X} \in \mathfrak{u}(V)$ and $\mathrm{Tr}_1$ is the partial trace over the first subspace. One can check that as expected $P_\mathfrak{g} d\phi(a) = d\phi(a)$.

## Proof of theorem 2.8

The following Lemma is fundamental to our main theorem, it may also be of independent interest. The proof can be found in the Supplementary Information.

**Lemma 4.1.** Let $G$ be a compact simple Lie group with Lie algebra $\mathfrak{g}$. Suppose $V$ is a finite-dimensional inner product space, $\phi : G \to \mathcal{U}(V)$ is a unitary representation of $G$, and $\mathbf{U}_g = \phi(g)$. In addition, $a \in \mathfrak{g}$, $\mathbf{A} = d\phi(a)$. Then the following holds:

$$\int_G (\mathbf{U}_g \mathbf{A} \mathbf{U}_g^\dagger)^{\otimes 2} \, dg = \frac{\|\mathbf{A}\|_\mathrm{F}^2}{d_\mathfrak{g}} \mathbf{K}. \tag{43}$$

From Lemma 4.1, it can be seen that the commutant is the one-dimensional subspace spanned by the Casimir operator, i.e.

$$\mathcal{T}(R) = \frac{\mathrm{Tr}(R^\dagger \mathbf{K})}{d_\mathfrak{g}} \mathbf{K} \quad \forall R \in \mathcal{S}. \tag{44}$$

We are also going to frequently use the following identity. Let $a \in \mathfrak{g}$ and $\mathbf{A} := d\phi(a)$. Also let $\mathbf{E}_i := d\phi(e_i)$ be a basis for the Lie algebra orthonormal under the standard norm. Then

$$\|\mathbf{A}\|_\mathrm{F}^2 = -I_\phi^{-1} \sum_i \mathrm{Tr}^2(\mathbf{A} \mathbf{E}_i), \tag{45}$$

which is important as when working with a quantum circuit one often has access to the representation basis $\{\mathbf{E}_i\}$ but not directly to $\{e_i\}$, so it is a convenient shortcut to calculate $\|a\|_\mathfrak{g}^2$.

**Proof of theorem 2.8.** As was shown in the Results section, we can assume $\mathrm{GradVar} = \mathbb{E}_{g^+, g^- \sim \mu^{\otimes 2}}[(\partial \langle \mathbf{O} \rangle_\rho)^2]$. Let us write the integral for the

second moment in full, and rearrange terms appropriately:

$$\mathbb{E}_{g^+, g^- \sim \mu^{\otimes 2}}[(\partial \langle \mathbf{O} \rangle_\rho)^2] \tag{46}$$

$$= \iint_G (\text{Tr}(\mathbf{U}_{g^-} i\rho \mathbf{U}_{g^-}^\dagger [\mathbf{H}, \mathbf{U}_{g^+} i\mathbf{O}\mathbf{U}_{g^+}^\dagger]))^2 \, dg^+ \, dg^- \tag{47}$$

$$= \iint_G \text{Tr}\left\{ (i\rho)^{\otimes 2} \, \mathbf{U}_{g^-}^{\otimes 2} \left( [\mathbf{H}, \mathbf{U}_{g^+} i\mathbf{O}\mathbf{U}_{g^+}^\dagger] \right. \right.$$
$$\left. \left. \otimes [\mathbf{H}, \mathbf{U}_{g^+} i\mathbf{O}\mathbf{U}_{g^+}^\dagger] \right) \mathbf{U}_{g^-}^{\dagger \otimes 2} \right\} \, dg^+ \, dg^-. \tag{48}$$

Suppose $\mathbf{X}_+ := \int_G (\mathbf{U}_{g^+} i\mathbf{O}\mathbf{U}_{g^+}^\dagger)^{\otimes 2} \, dg^+$. Let us ignore the trace and $\rho$, and expand out the commutators:

$$\iint_G \mathbf{U}_{g^-}^{\otimes 2} (\mathbf{H}\mathbf{U}_{g^+} i\mathbf{O}\mathbf{U}_{g^+}^\dagger - \mathbf{U}_{g^+} i\mathbf{O}\mathbf{U}_{g^+}^\dagger \mathbf{H})$$
$$\otimes (\mathbf{H}\mathbf{U}_{g^+} i\mathbf{O}\mathbf{U}_{g^+}^\dagger - \mathbf{U}_{g^+} i\mathbf{O}\mathbf{U}_{g^+}^\dagger \mathbf{H})\mathbf{U}_{g^-}^{\otimes 2} \, dg^+ \, dg^- \tag{49}$$

$$= \int_G \mathbf{U}_{g^-}^{\otimes 2} \mathbf{H}^{\otimes 2} \mathbf{X}_+ \mathbf{U}_{g^-}^{\dagger \otimes 2} \, dg^- \tag{50}$$

$$+ \int_G \mathbf{U}_{g^-}^{\otimes 2} \mathbf{X}_+ \mathbf{H}^{\otimes 2} \mathbf{U}_{g^-}^{\dagger \otimes 2} \, dg^- \tag{51}$$

$$- \int_G \mathbf{U}_{g^-}^{\otimes 2} (\mathbf{H} \otimes \mathbb{1}) \mathbf{X}_+ (\mathbb{1} \otimes \mathbf{H}) \mathbf{U}_{g^-}^{\dagger \otimes 2} \, dg^- \tag{52}$$

$$- \int_G \mathbf{U}_{g^-}^{\otimes 2} (\mathbb{1} \otimes \mathbf{H}) \mathbf{X}_+ (\mathbf{H} \otimes \mathbb{1}) \mathbf{U}_{g^-}^{\dagger \otimes 2} \, dg^-. \tag{53}$$

We end up with four similar terms. Starting with the common inner integral, since $G$ is compact, we can apply Lemma 4.1 and write

$$\mathbf{X}_+ = \int_G (\mathbf{U}_{g^+} i\mathbf{O}\mathbf{U}_{g^+}^\dagger)^{\otimes 2} \, dg^+ = \frac{\| \mathbf{O} \|_F^2}{d_{\mathfrak{g}}} \mathbf{K}. \tag{54}$$

We can plug this expression back into the earlier expression without the trace and $\rho$, and rearranging terms and using $\mathbf{K} := I_\phi^{-1} \sum_i \mathbf{E}_i \otimes \mathbf{E}_i$ gives:

$$\frac{\| \mathbf{O} \|_F^2}{I_\phi d_{\mathfrak{g}}} \sum_{k=1}^{d_{\mathfrak{g}}} \int_G \mathbf{U}_g [\mathbf{H}, \mathbf{E}_k] \mathbf{U}_g^\dagger \otimes \mathbf{U}_g [\mathbf{H}, \mathbf{E}_k] \mathbf{U}_g^\dagger \, dg. \tag{55}$$

Now applying the Lemma again, noting that $\mathbf{H} = \sum_q h_q \mathbf{E}_q$, we have:

$$\frac{\| \mathbf{O} \|_F^2}{I_\phi d_{\mathfrak{g}}^2} \sum_{j,k=1}^{d_{\mathfrak{g}}} \| [\mathbf{H}, \mathbf{E}_k] \|_F^2 \mathbf{K} \tag{56}$$

$$= \frac{\| \mathbf{O} \|_F^2}{I_\phi d_{\mathfrak{g}}^2} \sum_{j,k=1}^{d_{\mathfrak{g}}} \frac{\text{Tr}([\mathbf{H}, \mathbf{E}_k]\mathbf{E}_j)\text{Tr}([\mathbf{H}, \mathbf{E}_k]\mathbf{E}_j)}{I_\phi} \mathbf{K} \tag{57}$$

$$= \frac{\| \mathbf{O} \|_F^2}{d_{\mathfrak{g}}^2} \sum_{q,r,j,k=1}^{d_{\mathfrak{g}}} h_q h_r \frac{\text{Tr}([\mathbf{E}_q, \mathbf{E}_k]\mathbf{E}_j)\text{Tr}([\mathbf{E}_r, \mathbf{E}_k]\mathbf{E}_j)}{I_\phi^2} \mathbf{K} \tag{58}$$

$$= \frac{\| \mathbf{O} \|_F^2}{d_{\mathfrak{g}}^2} \sum_{q,r,j,k=1}^{d_{\mathfrak{g}}} h_q h_r f_{qk}^j f_{rk}^j \mathbf{K} \tag{59}$$

$$= \frac{\| \mathbf{O} \|_F^2}{d_{\mathfrak{g}}^2} \sum_{q,r=1}^{d_{\mathfrak{g}}} h_q h_r \left( -\sum_{j,k=1}^{d_{\mathfrak{g}}} f_{qk}^j f_{rj}^k \right) \mathbf{K} \tag{60}$$

$$= \frac{\| \mathbf{O} \|_F^2}{d_{\mathfrak{g}}^2} \sum_{q,r=1}^{d_{\mathfrak{g}}} h_q h_r \left( -g_{qr} \right) \mathbf{K} \tag{61}$$

$$= \frac{\| \mathbf{O} \|_F^2 \, \| \mathbf{H} \|_K^2}{d_{\mathfrak{g}}^2} \mathbf{K}, \tag{62}$$

where we have used anti-symmetry of the commutator braket to reveal that the inner sum is the Killing form (since $\mathfrak{g}$ is a compact simple Lie algebra, the negative of the Killing form is a valid inner product). Note that $f_{qk}^j = \text{Tr}([\mathbf{E}_q, \mathbf{E}_k]\mathbf{E}_j)$ are the structure constants.

Now, we can reintroduce the trace and $\rho$ to get:

$$\mathbb{E}_{g^+, g^- \sim \mu^{\otimes 2}}[(\partial \langle \mathbf{O} \rangle_\rho)^2] \tag{63}$$

$$= \frac{\| \mathbf{H} \|_K^2 \, \| \mathbf{O} \|_F^2}{d_{\mathfrak{g}}^2} \text{Tr}((i\rho)^{\otimes 2} \mathbf{K}) \tag{64}$$

$$= \frac{\| \mathbf{H} \|_K^2 \, \| \mathbf{O} \|_F^2 \, \| \rho_{\mathfrak{g}} \|_F^2}{d_{\mathfrak{g}}^2}. \tag{65}$$

## Proof of theorem 2.9

The following is a generalization of Lemma 4.1 to outside the simple group setting. The proof can be found in the Supplementary Information.

**Lemma 4.2**. Let $G$ be a compact and connected Lie group with Lie algebra $\mathfrak{g}$. Suppose $V$ is a finite-dimensional inner product space, $\phi : G \to \mathcal{U}(V)$ is a unitary representation of $G$, and $\mathbf{U}_g = \phi(g)$. In addition, $a \in \mathfrak{g}$, $\mathbf{A} = d\phi(a)$. Then the following holds:

$$\int_G (\mathbf{U}_g \mathbf{A} \mathbf{U}_g^\dagger)^{\otimes 2} dg = \sum_\alpha \frac{\| \mathbf{A}_{\mathfrak{g}_\alpha} \|_F^2}{d_{\mathfrak{g}_\alpha}} \mathbf{K}_{\mathfrak{g}_\alpha} + \mathbf{A}_c^{\otimes 2}, \tag{66}$$

where $\mathbf{A}_{\mathfrak{g}_\alpha}$ is the image of the component of $a$ in $\mathfrak{g}_\alpha$ under $d\phi$. Likewise, $\mathbf{K}_{\mathfrak{g}_\alpha}$ is the Casimir in the subalgebra $\mathfrak{g}_\alpha$.

The above result implies that we expect contributions to the variance from the various subalgebras. Indeed, the final expression for the variance is remarkably simple, since all the cross terms between different subalgebras vanish, and the abelian subalgebras do not contribute.

**Proof of theorem 2.9**. The proof largely follows the strategy of that for simple groups. Define the shorthand $i\mathbf{O}_{\mathfrak{g}_\alpha} := P_{\mathfrak{g}_\alpha} i\mathbf{O}$ and $i\mathbf{O}_c := P_c i\mathbf{O}$. Like before, we expand the commutator but this time use Lemma 4.2:

$$\int_G (\mathbf{U}_{g^+} i\mathbf{O}\mathbf{U}_{g^+}^\dagger)^{\otimes 2} dg^+ = \sum_\alpha \frac{\| \mathbf{O}_{\mathfrak{g}_\alpha} \|_F^2}{d_{\mathfrak{g}_\alpha}} \mathbf{K}_{\mathfrak{g}_\alpha} + \mathbf{O}_c^{\otimes 2}. \tag{67}$$

Now, after applying the commutator and taking the integral over $\mathbf{U}_{g^-}$, we find the result is still a summation over $\alpha$ only. This is because, since the subalgebras are ideals, if $\mathbf{E}_k \in d\phi(\mathfrak{g}_\alpha)$ then $[\mathbf{H}, \mathbf{E}_k] \in d\phi(\mathfrak{g}_\alpha)$, and therefore $\| P_{\mathfrak{g}_\beta}[\mathbf{H}, \mathbf{E}_k] \|_F = 0$ if $\beta \neq \alpha$. Thus the cross terms vanish. The contribution from the center also vanishes upon taking the commutator. Thus the result follows.

## Proof of theorem 2.11

Using the identities from the Representation theoretic notation subsection of Results we can get forms of the theorems that are practically useful. For example, in the simple group case,

$$\text{GradVar} = \frac{I_{\text{Ad}}\|o\|_{\mathfrak{g}}^2\|h\|_{\mathfrak{g}}^2}{d_{\mathfrak{g}}^2}\sum_i \text{Tr}^2(i\rho\mathbf{E}_i), \tag{68}$$

where $\mathbf{E}_i = d\phi(e_i)$ for orthonormal basis $\{e_i\}$ for $\mathfrak{g}$. This turns out to be the most useful form of the result for the examples below because we will have explicit knowledge of the representation $\phi$. In addition, the representation index, $I_\phi$, drops out.

**Proof of theorem 2.11.** For $\mathfrak{su}(n)$, $d_{\mathfrak{g}} = n^2 - 1$ and the Dynkin index of the adjoint representation is $I_{\text{Ad}} = 2n$. Now we work out the state's projected norm. Choose $\rho$ to be a computational basis state, where it can be shown that $\rho \otimes \rho$ lies in an irreducible subrepresentation of the tensor product representation $\phi \otimes \phi$ (see Supplementary Information). Then we only need to focus on the simultaneously diagonal elements of the Lie algebra, that is, the Cartan subalgebra $\mathfrak{h}$. To calculate the Casimir eigenvalue we need to find an orthogonal basis $\mathscr{H}$ for $\mathfrak{h}$, which cannot be $\{h_z^{ij}\}_{i\neq j}$ since the elements are not linearly independent.

We can construct a suitable basis for $\mathfrak{h}$ using the formula

$$\mathscr{H} = \frac{i}{2}\bigcup_{m=1}^{n-1}\frac{1}{\sqrt{m(m+1)}}\left\{m\sigma_{m+1}^z - \sum_{i=1}^m \sigma_j^z\right\} \tag{69}$$

$$= \frac{i}{4}\left\{\sqrt{2}(\sigma_2^z - \sigma_1^z), \frac{\sqrt{2}}{\sqrt{3}}(2\sigma_3^z - \sigma_2^z - \sigma_1^z),\right.$$
$$\left.\frac{1}{\sqrt{3}}(3\sigma_4^z - \sigma_3^z - \sigma_2^z - \sigma_1^z),\dots\right\} \tag{70}$$

even though this is expressed more cleanly with Pauli $z$s, each element can be obtained as a linear combination of the $\{h_z^{ij}\}$ generators. One can check that the elements are all orthogonal and the norm of their pullback on $\mathfrak{g}$ is 1, and the resulting subalgebra has the correct dimension: $\dim \mathfrak{h} = \text{rank } \mathfrak{su}(n) = n - 1$. With this, one can explicitly calculate the diagonal part of $I_\phi \mathbf{K}$ for any $n$,

$$I_\phi \text{diag}(\mathbf{K}) = \sum_{\mathbf{H}_i \in \mathscr{H}} \mathbf{H}_i \otimes \mathbf{H}_i \tag{71}$$

however the calculation is unwieldy. Fortunately, we can directly infer the final form from symmetry arguments, since by inspection: diag($\mathbf{K}$) is composed of sums of tensor products of two Pauli $z$s, it is symmetric around the tensor product, and furthermore since $\text{SWAP}_{ij} \otimes \text{SWAP}_{ij} \in \phi(G) \otimes \phi(G)$ it must be invariant upon any simultaneous permutation of the qubit indices on the subspaces. Thus,

$$I_\phi \text{diag}(\mathbf{K}) = A\sum_{i=1}^n \sigma_i^z \otimes \sigma_i^z + B\sum_{i\neq j}\sigma_i^z \otimes \sigma_j^z. \tag{72}$$

To find the value of $A$, evaluate diag($\mathbf{K}$) on the state $|\Psi\rangle = |+\dots+0\rangle^{\otimes 2}$ using Eqs. (71) and (72):

$$I_\phi\langle\Psi|\text{diag}(\mathbf{K})|\Psi\rangle = -\frac{1}{4n(n-1)}(n-1)^2\langle\Psi|\sigma_n^z \otimes \sigma_n^z|\Psi\rangle \tag{73}$$

$$= A\langle\Psi|\sigma_n^z \otimes \sigma_n^z|\Psi\rangle \;\Rightarrow\; A = -\frac{n-1}{4n}, \tag{74}$$

and for $B$, on $|\Psi'\rangle = |+\dots+0\rangle \otimes |+\dots+0+\rangle$:

$$I_\phi\langle\Psi'|\text{diag}(\mathbf{K})|\Psi'\rangle \tag{75}$$

$$= \frac{1}{4n(n-1)}(n-1)\langle\Psi'|(\sigma_n^z \otimes \sigma_{n-1}^z)|\Psi'\rangle \tag{76}$$

$$= B\langle\Psi'|\sigma_n^z \otimes \sigma_{n-1}^z|\Psi'\rangle \;\Rightarrow\; B = \frac{1}{4n}. \tag{77}$$

Now we use this to evaluate the expectation value of $\mathbf{K}$ on a computational basis state of Hamming weight $k$. The first summation in Eq. (72) will be constant and equal to $n$, while the second summation will be equal to the number of distinct bits of equal value minus those of different value, $k(k-1) + (n-k)(n-k-1) - 2k(n-k) = (n-2k)^2 - n$. So overall

$$I_\phi\|\rho_{\mathfrak{g}}\|_F^2 = \sum_{\mathbf{H}_i \in \mathscr{H}}\text{Tr}^2(i\rho\mathbf{H}_i) \tag{78}$$

$$= \frac{n-1}{4} - \frac{(n-2k)^2 - n}{4n} = \frac{k(n-k)}{n}. \tag{79}$$

Choosing $\mathbf{O} = ih_z^{12}$ and $\mathbf{H}$ any generator, $\|o\|_{\mathfrak{g}}^2 = 1/2 = \|h\|_{\mathfrak{g}}^2$, and the final result is

$$\text{GradVar} = \frac{2n(1/2)^2}{(n^2-1)^2}\frac{k(n-k)}{n} \tag{80}$$

$$= \frac{k(n-k)}{2(n^2-1)^2} \in \Omega\left(\frac{1}{n^3}\right). \tag{81}$$

**Proof of theorem 2.12.** For the uniform superposition of computational basis states, $|\psi\rangle = |+\rangle^{\otimes n}$, then $\forall i,j, \langle\psi|h_{ij}^y|\psi\rangle = \langle\psi|h_z^{ij}|\psi\rangle = 0$. The only nonzero terms involve the Pauli-$x$ type generators. We can form the corresponding orthogonal generators normalized in $\mathfrak{g}$ by $\mathbf{H}_{ij}^x = \sqrt{2}h_{ij}^x$. However, even though there are $\binom{0.0ptn}{2}$, only the $n-1$ with $j = i+1$ do not annihilate on $|\psi\rangle$ since the others have $\sigma_z$'s in their definition. For these generators, $\langle\psi|\mathbf{H}_{ij}^x|\psi\rangle = -\frac{i}{2\sqrt{2}}$, giving

$$I_\phi\|P_{\mathfrak{g}}\rho\|_F^2 = -\sum_{i=1}^{n-1}|\langle\psi|\mathbf{H}_{i(i+1)}^x|\psi\rangle|^2 = \frac{1}{8}(n-1), \tag{82}$$

and so

$$\text{GradVar} = \frac{2n(1/2)^2}{(n^2-1)^2}\frac{(n-1)}{8} = \frac{n(n-1)}{16(n^2-1)^2} \in \Theta\left(\frac{1}{n^2}\right). \tag{83}$$

**Proof of corollary 2.12.1.** We expand the variance term for the computational basis state case:

$$\frac{2k(n-k)}{(n^2-1)^2} = \text{Var}_{(g_+, g_-)\sim\mu^{\otimes 2}}[\partial\langle\sigma_i^z - \sigma_j^z\rangle] \tag{84}$$

$$= \text{Var}_{(g_+, g_-)\sim\mu^{\otimes 2}}[\partial\langle\sigma_i^z\rangle] + \text{Var}_{(g_+, g_-)\sim\mu^{\otimes 2}}[\partial\langle\sigma_j^z\rangle] \tag{85}$$

$$- 2\text{Cov}_{(g_+, g_-)\sim\mu^{\otimes 2}}[\partial\langle\sigma_i^z\rangle, \partial\langle\sigma_j^z\rangle]. \tag{86}$$

Note since a permutation swapping qubit $i$ with $j$ is a valid compound SU matrix, we have that $\partial\langle\sigma_i^z\rangle$ and $\partial\langle\sigma_j^z\rangle$ are identically distributed. Thus,

$$\begin{aligned}\frac{2k(n-k)}{(n^2-1)^2} &= 2\text{Var}_{(g_+, g_-)\sim\mu^{\otimes 2}}[\partial\langle\sigma_i^z\rangle]\\ &- 2\text{Cov}_{(g_+, g_-)\sim\mu^{\otimes 2}}[\partial\langle\sigma_i^z\rangle, \partial\langle\sigma_j^z\rangle].\end{aligned} \tag{87}$$

Due to the above equality and Cauchy–Schwarz, i.e., $\mathrm{Var}_{(g_+, g_-) \sim \mu^{\otimes 2}}[\partial \langle \sigma_i^z \rangle] \geq |\mathrm{Cov}_{(g_+, g_-) \sim \mu^{\otimes 2}}[\partial \langle \sigma_i^z \rangle, \partial \langle \sigma_j^z \rangle]|$ (recall the variances are equal), we can conclude that $\mathrm{Var}_{(g_+, g_-) \sim \mu^{\otimes 2}}[\partial \langle \sigma_i^z \rangle]$ must only be polynomially vanishing in $n$, which implies no BP for any $k$ and any single-qubit $\sigma_z$ measurement. A similar result can be shown to hold for the uniform superposition state.

## Data availability
The gradient variance simulation data generated in this study have been deposited in the Zenodo database under accession code https://doi.org/10.5281/zenodo.10720106.

## Code availability
The code used to generate the gradient variance simulation data has been deposited in the Zenodo database under accession code https://doi.org/10.5281/zenodo.10720106.

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

## Acknowledgements

We thank Iordanis Kerenidis for early discussions on BPs in quantum compound ansätze, and Aram Harrow for helpful discussions and feedback on the manuscript. We thank Marco Cerezo and Martin Larocca for discussions on the basics of equivariant QNNs and the role of the DLA. We also thank the members of Global Technology Applied Research at JPMorgan Chase & Co. for their comments and feedback throughout the project. This paper was prepared for informational purposes by the Global Technology Applied Research Center of JPMorgan Chase & Co. This paper is not a product of the Research Department of JPMorgan Chase & Co. or its affiliates. Neither JPMorgan Chase & Co. nor any of its affiliates make any explicit or implied representation or warranty, and none of them accept any liability in connection with this paper, including, without limitation, with respect to the completeness, accuracy, or reliability of the information contained herein and the potential legal, compliance, tax, or accounting effects thereof. This document is not intended as investment research or investment advice, or as a recommendation, offer, or solicitation for the purchase or sale of any security, financial instrument, financial product, or service, or to be used in any way for evaluating the merits of participating in any transaction.

## Author contributions

D.H. and S.C. conceived the research question, and E.F. wrote the first proof of Conjecture 2.3, to which D.H. and S.C. made significant improvements. E.F. wrote the proof to Theorem 2.11, 2.12, and Corollary 2.12.1. D.H. wrote the proof for Theorem 2.6, 2.10, 2.13, and the majority of the theory in the Supplementary Information, with contributions from S.C. N.K., R.Y., J.H., S.H.S., and M.P. contributed to the technical discussions and had a role in writing and proof-reading the manuscript.

## Competing interests

The authors declare no competing interests.
