## [Peer Review File · Nature Communications]

Characterizing Barren Plateaus in Quantum Ansätze with the Adjoint RepresentationREVIEWER COMMENTS

Reviewer #1 (Remarks to the Author):

The authors study the barren plateau phenomenon, which is the generic concentration of gradients in variational quantum algorithms. It previously was conjectured that the variance of these gradients scaled inverse polynomially with the dimension of the so-called "dynamical Lie algebra," which is the Lie algebra generated by the generators of the unitary evolution of the parameterized quantum circuit (in the subspace the initial state is in). The authors here prove a stronger version of this conjecture, exactly calculating the variance of gradients (i.e., not just upper-bounding it). They achieve this via some impressive algebraic techniques. The presented results tie up loose ends in understanding the loss landscapes of variational quantum algorithms, and I recommend this work for publication after only some minor edits.

First, I would recommend the authors explain the origin of Eq. (23). How does this "abstract gradient" relate to gradients with respect to parameters in certain simple cases? An example or two showing how this relates to the case where, e.g., $\mathfrak{g}=\mathfrak{su}(N)$ would be helpful, as well as some intuition as to why this should generally be the quantity we care about when evaluating whether or not barren plateaus are present in variational quantum algorithms.

Second, the numerical experiments performed by the authors are nowhere described in detail. For Fig. 4 they are not even described in the caption. An Appendix dedicated to explaining how these figures were generated is important for reproducibility. The figures also lack empirical error bars, which are important when evaluating gradient scaling via just random sampling of points on the landscape.

Finally, I would recommend moving many of the details of the proofs in the Main Result section to a Methods section. In my opinion this would greatly aid in the readability of the manuscript (while also fitting Nature Communication's typical formatting).

Reviewer #2 (Remarks to the Author):

Disclaimer: Given time limitations, I was asked to write a short review of this paper so I did not go through the technical details or check for mathematical correctness. I also did not read all parts of this paper in the typical detail reserved for such reviews, and there may be important things I have missed in writing this review.

The authors study the barren plateau problem for variational quantum circuits constrained under given symmetries. Their results provide a direct means to analyze the variance of gradients when the structure of the dynamical Lie algebra is known and circuits are assumed to be initialized as a two design. These formulas are written in terms of the dimensions and projections of given observables and inputs onto the dynamical Lie algebra. Various examples are later provided that I thought were nice to read and understand.

At a high level, there are a few weaknesses of this paper that I see:

- The authors assume a 2-design in their proofs which sidesteps the perhaps more practical question which is “how fast does a given circuit actually approach a two design?” This is the main question that practitioners face and largely unaddressed here.
- Many of the parameters in the main theorem can be hard to evaluate. E.g. dimensions of the simple ideals or projections of a given hamiltonian onto a symmetric subspace may not be easy to calculate. Nonetheless, getting a sense of whether it is exponentially large or polynomially large may be feasible in general.
- Practical quantum architectures constrained by symmetries may have strong limitations on controllability that are largely unaccounted for in this two-design assumption (see [3]). Also, practical systems typically are not fully constrained by symmetries but approximately constrained (e.g. as in classical convolutional networks) which can change the story significantly.

Overall, the contribution is a meaningful one to the quantum community, and this work takes a notable step forward in understanding learnability of variational algorithms in the symmetric setting. Given the weaknesses above and the rather specific setting the authors study, I am unsure of the study's relevance to the broader community. I would definitely

support publication of this paper in a quantum focused journal, but whether the results are novel enough for a broad audience is a question I leave up to the editor.

Smaller comments:

- Definition 1.2 has a technical subtlety that can be misleading. The authors define a barren plateau as one where the variance decays as $O(1/2^n)$, but this is a bit strict since it requires that the decay be at least linearly exponential in n . A more accurate criteria would say that the variance decays as $O(2^{-\Omega(n)})$ or equivalently $O(2^{-cn})$ for some $c > 0$.
- Notation and definition in 2A is not clearly defined, E.g. V_{ϕ} , \mathfrak{u} , etc. Some of these can be inferred, but the authors should state what they are explicitly nonetheless.
- The second sentence says "VQAs have found a variety of applications in the areas of..." This is a bit misleading. VQAs have yet to find any applications since we do not have any quantum computers yet to use them for and any algorithms so far are yet to have useful provable guarantees. Instead, the authors should say something like "VQAs are a promising potential application..." or something of that sort.
- The authors later state "It has been observed that both of these obstacles to VQA optimization can be mitigated when the chosen parameterized quantum circuit (PQC) obeys symmetries". Again, this depends on the symmetry and this statement must be qualified. This is only true for very specific symmetries where the symmetry is strong enough. E.g., see [1-2].

As a side note, and at no fault to the authors at all, the results in this paper are similar to those in a recent preprint by Ragone et al. titled "A Unified Theory of Barren Plateaus for Deep Parametrized Quantum Circuits". The main theorems are very similar. The main difference appears to be the more representation theoretic focus of the present work. This work also presents some examples that helped give more This similarity should not be held against the authors since it appears the two papers were concurrent with each other.

1. Anschuetz, Eric R., et al. "Efficient classical algorithms for simulating symmetric quantum systems." arXiv preprint arXiv:2211.16998 (2022).

2. Schatzki, Louis, et al. "Theoretical guarantees for permutation-equivariant quantum neural networks." arXiv preprint arXiv:2210.09974 (2022).

3. Marvian, Iman. "Restrictions on realizable unitary operations imposed by symmetry and locality." Nature Physics 18.3 (2022): 283-289.

Reviewer #3 (Remarks to the Author):

see pdf attached

[**Editorial note:** Please see the following pages for Reviewer #3's report.]

REVIEW: THE ADJOINT IS ALL YOU NEED

General Observation

This paper offers very strong results helping the community to understand the link between trainability and controllability using a framework (the LASA hypothesis) that is relevant for many quantum algorithms. The mathematical proofs are detailed and explained, and the motivation and intuition of the authors are shared with the readers, offering very valuable inputs. However, the article struggle to present in a clear way the results and to make the link with the community and other works. All information to do so are in the article, but the overall structure is not the most relevant to affect the community.

In addition, this paper came out simultaneously with another couple of papers arXiv:2309.09342 + arXiv:2310.11505, claiming some similar results, using similar proofs. In the reviewed paper, some aspects are better (maths details) but some are worst (wording, presentation, figures).

Abstract

Overall, the abstract sums up the contribution of the paper in a clear way. However, two sentences are misleading:

- “In particular, our theory provides for the first time the ability to compute the variance of the gradient of the cost function for a non- trivial, subspace uncontrollable family of quantum circuits, the quantum compound ansätze.”: First, it may be not clear what the authors mean by ”subspace uncontrollable family of quantum circuits”. Variance of the cost function have been study even in the case of non full controllability (for example: arXiv:2011.02966). As the notion of subspace preserving VQC is not clearly explained or very common in the community, it could be misleading. In addition, calling ”quantum compound ansätze” the use of Hamming Weight preserving VQC is a mistake. In Section IV-B, the authors define the Hamming Weight preserving VQC and cite the example of the use of RBS agtes, but it has been established that RBS based circuits are not always compound ansatzes (see arXiv:2202.00054, arXiv:2209.08167 & arXiv:2309.15547). This article also gives a way to derive the variance of the cost function for Hamming Weight preserving ansatzes, using different mathematical tools.
- ”We rigorously prove that the variance of the gradient of the cost function, under Haar initialization, scales inversely with the dimension of the DLA, which agrees with existing numerical observations.”: it is not clear that the authors prove this result in the case of LASA.

1 Introduction

The introduction states very well the motivation of the authors, and I believe that it is very valuable that the authors made the effort to explain where their intuition came from. The explanation are very clear and mathematically very accurate, but the paper could improve it simplicity to reach a larger audience. I believe it is very important due to the fact that the main result linking controllability and trainability considering the LASA framework will impact the all Quantum Machine Learning community.

- “The symmetries of the ansatz cause its action to break into invariant subspaces, and in each invariant subspace the quantities controlling trainability and convergence only depend on characteristics of the subspace, e.g., its dimension.”: This is a bit misleading. The authors actually prove this result in the case of LASA but it was just a conjecture before this and it is still false beyond LASA (as you show in Appendix C, and in arXiv:2310.11505, arXiv:2309.15547).
- “However, it is challenging in general to determine if subspace controllability can be attained, and there are commonly used PQCs that do not attain it, such as the quantum compound ansatz”: same issue as before with

this term. In addition, RBS-based seems to be able to reach full controlability but FBS (Fermionic Beam Splitter - see [arXiv:2202.00054](https://arxiv.org/abs/2202.00054)) cannot, due to the fact that FBS are indeed quantum compound ansatz (not RBS).

- Even if the authors explain in a very clear way why the link between the dimension of the Lie Algebra and the trainability is not trivial, I believed that it could be helpful to refer to existing well-known work. For example, the authors present in Equation (6) the second-moment integral operator which is linked with the variance of the gradient. While it is absolutely true and mathematically clear, it could be very helpful to refer to other papers that make the link between 2-design and Barren Plateau.
- From Equation (6) to Definition I.5, the authors give precious intuition to explain how they had the idea of defining the LASA framework. However, I am afraid that it is a bit too much to digest for the reader. In addition, it could be good to give a simple and short definition of the adjoint representation to increase the pedagogical aspect of the paper.
- Figure 1: I believe the use of Figure will help to improve the clarity of the work, but Figure 1 is not very useful. It only helps the reader to understand that the observable and the Hamiltonians are part of the Lie algebra (and it is not that much clear that the star is the observable), which was already clear in the text.
- In Equation (5) both norms $\|H\|_K$ and $\|H\|_F$ are used, is it on purpose?

2 Mathematical Preliminaries

2.1 Review of representation theory

In this subsection, the authors present the tools they used from the representation theory in order to help the reader to understand the tools used to obtain their main results. However, I am not sure to who this part is addressed. It is very unclear for people that are not familiar with representation theory, and too many concepts and notions are called but not explained. I will advise to give a more low-level presentation of the tools they use and to give in appendix an extended version of this review.

- "Consider a complex, finite-dimensional unitary representation $\phi : G \rightarrow U(V_\phi)$ ": not very clear as V_ϕ not defined (is it the reduced Hilbert space?). Same for Tr_ϕ, U_g .
- The motivation of giving Equation (13) is unclear. This equation is used in the Section IV-A, but it is not clear why it is define and useful.
- It is puzzling that the normalized split Casimir invariant (or operator) is presented and motivated in a review and that the authors advise the reader to compare it with "the usual Casimir operator". If the goal of this section is to introduce basic concepts and mathematical tools, it should not rely on a deep knowledge from the reader (or it should be placed in appendix).

This part is essential for the comprehension of the rest of the paper, and we acknowledge that it must be hard to balance between the strict minimum and a full course on Lie Theory. But at least each term and notation used must be properly define somewhere, even if they are very standard for some mathematicians in this field.

2.2 Representation-theoretic framework for barren plateaus

This part is more clear than the previous one, the motivation is well presented in the beginning.

- Definition II.1: "where $U_{g^\pm} = \phi(g^\pm)$ for arbitrary $g_+, g_- \in G$. It seems like g_+, g_- are arbitrary elements of G but equation (24) and the following definition of $U(\Theta)$ seems to indicate that the signs \pm indicate part of the circuits. It should be clarify in the Definition II.1.

3 Main Result

The Theorems and Lemmas are well states, and the critics of mathematical tools used are well explained.

- Theorem III.2: comments on the importance of the terms $\|H\|_K^2, \|O\|_K^2,$ and $\|P_g \rho\|_K^2$ would be appreciated. In other work ([arXiv:2309.09342](https://arxiv.org/abs/2309.09342)) the notion of purity of hermitian operator is introduced and discuss, and I think it is important to also discuss it here. It is discussed in subsections A and B but it could be helpful to have already some comments and a link with this subsection.

- Definition III.3: to improve the clarity, the authors should make the link with other works using reducible Lie Algebra (arXiv:2105.14377, arXiv:2202.00054, ...) and explained again the link with symmetries and subspace preserving ansatzes.
- Theorem III.5: this result is only introduced by the paragraph just before. It is important to add comments on how it affect quantum ansatzes using reducible Lie Algebra in practice.
- Theorem III.6: here again, the impact of this theorem is not discussed directly. In addition, the statement of the Theorem should be straightforward "consider a vector $|v\rangle$ in an irrep V " is not clear for example.
- "C. Connection with existing approach": First some terms must be defined " ϕ is faithful", or " ϕ is not faithful" (it was already mentioned but not defined). The connection with the setting of considering subrepresentations is not very clear, especially this sentence "While it may seem that the if the initial state has support across multiple subrepresentations that the variance splits into a sum across the subrepresentations, this is not the case. This is simply because of covariance terms that are not zero in general, i.e. the Haar measure does not always factor across subrepresentations".

4 Examples

We don't think it is a good idea to present the case uncontrollable systems as just examples. One of the main question before this article was to understand the scaling of the gradient when considering non controllable systems. It should be mentioned with the main theorems.

In addition, the authors should take the time to explained what happens when considering a circuit, with its corresponding Lie Group (that could be $su(d)$), but which is not parametrized enough to reach all unitaries in the Lie Group. This is especially important to avoid any confusion on the meaning of "full controllability".

We insist: RBS based circuits are NOT always compound ansatz. Only FBS are. And RBS = FBS only when qubits are connected to their nearest neighbors (arXiv:2202.00054, arXiv:2209.08167 & arXiv:2309.15547).

5 General Observables

This section does not have a clear introduction. The authors explained that "we need to allow for observables that have support outside the Lie algebra", but the content of this section should be summarize at the beginning. A good summary is done at the very last paragraph, but the link with the rest of the section is unclear. It is also very hard to understand Theorem V.1 and the link with the conclusion.

6 Review Conclusion

We believe this paper is a key contributions to the field. The level of mathematical details is impressive. It is a great paper and we recommend it for publication once the above mathematical explanations, wording clarification, and missing references are fixed.

Response to the Reviewers

Re: Manuscript ID NCOMMS-23-49122

“The Adjoint Is All You Need:
Characterizing Barren Plateaus in Quantum Ansätze”

Enrico Fontana, Dylan Herman, Shouvanik Chakrabarti, Niraj Kumar, Romina Yalovetzky, Jamie Heredge,
Shree Hari Sureshbabu, Marco Pistoia

Dear Reviewers,

We would like to thank you for your assessment of our work and providing critical feedback, which we have used to improve the presentation. Enclosed we submit our revised manuscript. The changes to the manuscript that directly answer comments or we would like you draw attention to are indicated in blue font. Below we address the comments individually.

We would like to highlight that we have modified the format of the manuscript to fit the Nature Communications styling. Thus, the manuscript will appear to have changed significantly, even though the content is largely the same. At the same time, this has resulted in a significantly clearer presentation of our results, and we hope that it is now more suitable for a wider audience. In particular, we made our best effort to outline the key results and their proofs in the clearest and most concise way. We also placed all additional results of a more theoretical interest in the *Supplementary Information*, alongside with lemmas and proofs which we judged too technical. We hope these changes are satisfactory.

Thank you,

Enrico Fontana on behalf of all authors.

Reviewer #1, comment #1

First, I would recommend the authors explain the origin of Eq. (23). How does this “abstract gradient” relate to gradients with respect to parameters in certain simple cases? An example or two showing how this relates to the case where, e.g., $\mathfrak{g} = \mathfrak{su}(N)$ would be helpful, as well as some intuition as to why this should generally be the quantity we care about when evaluating whether or not barren plateaus are present in variational quantum algorithms.

Our response #1.1

We appreciate the Reviewer bringing this to our attention. Indeed in the original version the ‘abstract gradient’ was poorly motivated. We have clarified the connection between ‘abstract gradient’ and the usual PQC gradient and what are the required assumptions for them to be related. Notably, these assumptions are valid for the generic periodic ansatz that we assumed, when the number of layers is deep enough to guarantee convergence (see Theorem 2.6). These details have been added after Definition 2.5.

Reviewer #1, comment #2

Second, the numerical experiments performed by the authors are nowhere described in detail. For Fig. 4 they are not even described in the caption. An Appendix dedicated to explaining how these figures were generated is important for reproducibility. The figures also lack empirical error bars, which are important when evaluating gradient scaling via just random sampling of points on the landscape.

Our response #1.2

We concur with the Reviewer that the instructions for reproducing the Figures were insufficient, and we thank the Reviewer for highlighting this. We now include explicit instructions on how to reproduce the

experimental results in “Details of Numerical Results” section of the Supplementary Information. In the caption of all the figures we refer to this section.

Regarding the error bars, it appears that the standard error is too small to even be noticeable on the graphs. Each data point is an unbiased estimate of the variance and for most data points we have taken around 5000 samples (uniformly initializations of circuit parameters). This is in line with previous experiments like those in DOI:q-2022-09-29-824/, where no error bars were displayed either. Specifically, it appears that the convergence to approximate 2-design is very fast and thus the statistical error when estimating the variance is also small.

Reviewer #1, comment #3

Finally, I would recommend moving many of the details of the proofs in the Main Result section to a Methods section. In my opinion this would greatly aid in the readability of the manuscript (while also fitting Nature Communication’s typical formatting).

Our response #1.3

We agree with the Reviewer on this important point. We realize that the format of the paper was not fit for publication, and too many details were a barrier to readability. In hindsight it would have been preferable to reformat before submission.

The paper has now been reworked to fit the Nature Communications’ format. The new Results section now only presents the theorems and discusses the impact with no proofs included. The discussion of representation-theoretical concepts in the main text is kept at a minimum. In addition, the most important proofs have been moved to the Methods section, while the rest are now in the Supplementary Information.

Reviewer #2, comment #1

The authors assume a 2-design in their proofs which sidesteps the perhaps more practical question which is “how fast does a given circuit actually approach a two design?” This is the main question that practitioners face and largely unaddressed here.

Our response #2.1

This is a valid concern, and we thank the Reviewer for highlighting it. Indeed the question of convergence to 2-design is crucial to the applicability of the results and should be thoroughly addressed.

We had added a new result (Theorem 2.6) showing that for LASA with polynomial DLA, convergence to a 2-design for the dynamical group is very rapid. This result is based on very recent work on convergence of random Pauli rotations to 2-designs, and is significantly stronger than previous ones which ignore the spectral gap. In addition, we provide a tighter analysis for the quantum compound ansatz (Theorem 2.10) that matches the scaling of the number of layers utilized in the numerics.

A more general result for arbitrary LASA and t -design has been added to the Supplementary Information section. The mixing rate depends on a quantity that we call the minimum stable Killing rank and varies with the choice of generators. The mixing to a design is faster the closer this rank is to the dimension of the algebra.

These are all now highlighted as new contributions in the abstract, and we really appreciate the reviewer for pointing out this issue.

Reviewer #2, comment #2

Many of the parameters in the main theorem can be hard to evaluate. E.g. dimensions of the simple ideals or projections of a given hamiltonian onto a symmetric subspace may not be easy to calculate. Nonetheless, getting a sense of whether it is exponentially large or polynomially large may be feasible in general.

Our response #2.2

We appreciate the Reviewer bringing up this point. Indeed the results were not framed in a sufficiently practical form. We have since thought more about how one would in practice compute the quantities in our formulae efficiently. It appears that when the DLA is polynomial, one can do this. Note that polynomial

DLA does not imply no BP (since the projected norm of the state on the DLA may still be vanishing), so computing these quantities is still important.

There are now two paragraphs in the new Section 2 E (Interpretation of Results) sketching how one could efficiently compute these quantities in practice. Thus for polynomially sized DLA, which is the case of interest for LASA, these quantities are *not* hard to compute. A quantum computer is only needed for evaluating the projected norm of the input state, which is an unavoidable step for generic input states.

Reviewer #2, comment #3

Practical quantum architectures constrained by symmetries may have strong limitations on controllability that are largely unaccounted for in this two-design assumption (see [3]). Also, practical systems typically are not fully constrained by symmetries but approximately constrained (e.g. as in classical convolutional networks) which can change the story significantly.

Our response #2.3

The Reviewer brings up some interesting points that merit consideration. For the first point, indeed it may be that due to some limitation in the gateset (eg. in the reference provided the constraint is the gates' locality) the PQC may be unable to fully express some group. However we would like to explain how this still falls within our setting. For our analysis to hold, it is sufficient to assume convergence to an approximate 2-design of some compact Lie group, which is the case even in the presence of constraints. The key point is that indeed this group may not be the full unitary group on the subspace. This is what we refer to as *subspace uncontrollable* setting, and it is arguably the main contribution of the paper to show that the gradient variance may be calculated in this setting too.

In the new version of the manuscript, we have added more emphasis on the fact that our results hold regardless of controllability. Specifically, in Section 2 G (Comparison with Previous Approaches) we compare our approach with previous ones, and explain how the shift from Schrödinger to Heisenberg picture is key to overcoming the problems posed by uncontrollability.

Regarding the second point, about what happens when the symmetries are only approximate, this is a very interesting question. We would like to remark that in the quantum case one would always converge to some Lie group, at worst the full unitary group $U(2^n)$. At the same time, a symmetry may be slightly broken, such that at polynomial depth the circuit approximately mixes to a subgroup H of the true dynamical Lie group G . In that case however, as long as there is a sufficiently good convergence to a 2-design of H , the results will be valid using H instead of G . In addition, in Theorem 2.6 we now show that for generators randomly sampled from the DLA basis convergence is polynomial in depth, and so at least in this setting the problem does not occur. It is an interesting question to study settings with slight symmetry breaking, which we leave to future work.

Reviewer #2, comment #4

Definition 1.2 has a technical subtlety that can be misleading. The authors define a barren plateau as one where the variance decays as $O(1/2^n)$, but this is a bit strict since it requires that the decay be at least linearly exponential in n . A more accurate criteria would say that the variance decays as $O(2^{-\Omega(n)})$ or equivalently $O(2^{-cn})$ for some $c > 0$.

Our response #2.4

We thank the Reviewer for spotting this mistake, indeed the Reviewer is absolutely correct. We have updated the definition to account for other rates of exponential decay.

Reviewer #2, comment #5

Notation and definition in 2A is not clearly defined, E.g. V_ϕ, \mathbf{u} , etc. Some of these can be inferred, but the authors should state what they are explicitly nonetheless.

Our response #2.5

The observation is correct, and we apologize for the confusion. We have now added definitions for $\mathbf{u}(V)$, $U(V)$, tensor product of representations. We have mostly removed the representation subscript from the vector spaces when it is obvious which representation we are referring to. As a result all V_ϕ are now just V .

All the definitions required for understanding the setting and the results are now outlined in Section 2 A, 2 B, and 2 C.

Reviewer #2, comment #6

The second sentence says "VQAs have found a variety of applications in the areas of. . . " This is a bit misleading. VQAs have yet to find any applications since we do not have any quantum computers yet to use them for and any algorithms so far are yet to have useful provable guarantees. Instead, the authors should say something like "VQAs are a promising potential application. . ." or something of that sort.

Our response #2.6

We agree with the Reviewer that this is an important clarification to add. The change has been made to the intro. We changed the sentence highlighted to emphasize that there is only potential application and no actual application currently.

Reviewer #2, comment #7

The authors later state "It has been observed that both of these obstacles to VQA optimization can be mitigated when the chosen parameterized quantum circuit (PQC) obeys symmetries". Again, this depends on the symmetry and this statement must be qualified. This is only true for very specific symmetries where the symmetry is strong enough. E.g., see [1-2].

Our response #2.7

We agree that this statement requires more clarification as the notion of what symmetries are required for existing results to apply is unclear. We now emphasize that there are two notions of symmetries and hence invariant subspaces, those of the initial state (Schrödinger picture) and those of the observable (Heisenberg picture). Prior results have only theoretically proven that when the state lies in an invariant subspace (hence Schrödinger picture) and we have subspace controllability then the properties of trainability and convergence are characterized by this subspace.

We note that throughout the paper we use the notion of symmetry in the more general sense of any unitary operator or Hermitian that commutes with the whole group, and may not have any practical significance. We agree that it is important to emphasize that only a few useful cases have been identified (e.g. permutation invariance), which we believe is what the Reviewer is emphasizing. There is now a sentence stating this. Sentences clarifying the situation can now be found in the Introduction, as well as an in-depth discussion in Section 2 G.

Abstract

Reviewer #3, comment #1

"In particular, our theory provides for the first time the ability to compute the variance of the gradient of the cost function for a non-trivial, subspace uncontrollable family of quantum circuits, the quantum compound ansätze": First, it may be not clear what the authors mean by "subspace uncontrollable family of quantum circuits". Variance of the cost function have been study even in the case of non full controllability (for example: arXiv:2011.02966). As the notion of subspace preserving VQC is not clearly explained or very common in the community, it could be misleading.

Our response #3.1

We agree with the Reviewer that the highlighted statement can be misleading and the term 'subspace uncontrollable' may not be known to the broader community. The Reviewer is also correct in saying that the variance has been studied in uncontrollable settings, however, we remark that this was possible only for very specific ansätze.

The highlighted sentence has been changed to say that this enables the computation of the variance of the gradient for a specific family of quantum circuits, specifically the quantum compound ansatz. In

addition, throughout the paper we have added further details around the concept of subspace controllability and how our approach gets around uncontrollability, since we now realize this was a major shortfall of our presentation.

Reviewer #3, comment #2

In addition, calling "quantum compound ansatz" the use of Hamming Weight preserving VQC is a mistake. In Section IV-B, the authors define the Hamming Weight preserving VQC and cite the example of the use of RBS gates, but it has been established that RBS based circuits are not always compound ansatzes (see arXiv:2202.00054, arXiv:2209.08167 & arXiv:2309.15547).

This article also gives a way to derive the variance of the cost function for Hamming Weight preserving ansatzes, using different mathematical tools.

Our response #3.2

We agree with the Reviewer that the mention of RBS gates is not correct and have removed it. We also agree that compound ansätze are just a subset of Hamming weight preserving ansätze. We now emphasize that our focus is on the quantum compound ansatz, which utilizes generalized Givens rotations and are implemented with FBS gates only. We also now cite arXiv:2309.15547 in Section 2 F.

Reviewer #3, comment #3

"We rigorously prove that the variance of the gradient of the cost function, under Haar initialization, scales inversely with the dimension of the DLA, which agrees with existing numerical observations.": it is not clear that the authors prove this result in the case of LASA.

Our response #3.3

We thank the Reviewer for spotting this oversight on our end. We have clarified that our results are for LASAs and have updated the abstract.

Introduction

Reviewer #3, comment #4

The introduction states very well the motivation of the authors, and I believe that it is very valuable that the authors made the effort to explain where their intuition came from. The explanation are very clear and mathematically ver accurate, but the paper could improve it simplicity to reach a larger audience. I believe it is very important due to th fact that the main result linking controlability and trainability considering the LASA framework will impact the al Quantum Machine Learning community.

Our response #3.4

We thank the Reviewer for the kind comments on the accuracy and impact of the paper. We recognize that parts of the manuscript may have been inaccessible to many in the QML community. Even though the final result is quite intuitive, indeed many intermediate steps of the proof are technical, and in the original version we opted to be faithful to our research process.

We have now considerably revised the structure of the manuscript, such that the main text presents the results clearly and is sufficient to understand the proof technique, but does overburden the reader with details. The technicalities have been moved to the Supplementary Information.

Reviewer #3, comment #5

"The symmetries of the ansatz cause its action to break into invariant subspaces, and in each invariant subspace the quantities controlling trainability and convergence only depend on characteristics of the subspace, e.g., its dimension.": This is a bit misleading. The authors actually prove this result in the case of LASA but it was just a conjecture before this and it is still false beyond LASA (as you show in Appendix C, and in arXiv:2310.11505, arXiv:2309.15547).

Our response #3.5

We appreciate the Reviewer bringing this to our attention and agree that this statement about prior

results is not exact and requires clarification. Indeed prior theoretical results for both trainability and convergence exist only for the subspace controllable setting. We now emphasize that this falls in the Schrödinger picture. and that the invariant subspace controlling these properties in the LASA setting requires the Heisenberg picture (hence the adjoint representation). We have also added a statement emphasizing that we will introduce sufficient conditions on the observable for the DLA-BP connection to hold, i.e. LASA.

Reviewer #3, comment #6

“However, it is challenging in general to determine if subspace controllability can be attained, and there are commonly used PQCs that do not attain it, such as the quantum compound ansatz”: same issue as before with this term. In addition, RBS-based seems to be able to reach full controlability but FBS (Fermionic Beam Splitter - see arXiv:2202.00054) cannot, due to the fact that FBS are indeed quantum compound ansatz (not RBS).

Our response #3.6

We agree that the sentence highlighted by the Reviewer may be a source of confusion for the reader. Since it is not crucial to the discussion, we have decided to remove it.

Reviewer #3, comment #7

Even if the authors explain in a very clear way why the link between the dimension of the Lie Algebra and the trainability is not trivial, I believed that it could be helpful to refer to existing well-known work. For example, the authors present in Equation (6) the second-moment integral operator which is linked with the variance of the gradient. While it is absolutely true and mathematically clear, it could be very helpful to refer to other papers that make the link between 2-design and Barren Plateau.

Our response #3.7

We thank the Reviewer for this helpful suggestion. We decided that it would make the paper clearer to move the discussion on the second moment integral to the beginning of the new Methods section. There we also refer to prior work on the connection between second moment integrals, 2-designs and BPs.

Reviewer #3, comment #8

From Equation (6) to Definition I.5, the authors give precious intuition to explain how they had the idea of defining the LASA framework. However, I am afraid that it is a bit too much to digest for the reader. In addition, it could be good to give a simple and short definition of the adjoint representation to increase the pedagogical aspect of the paper.

Our response #3.8

We agree with the Reviewer that the original introduction was too technical. The discussion highlighted by the Reviewer has been moved to the Methods section. Regarding the excellent suggestion of giving a short definition of the adjoint representation, this has been implemented in Section 2 C (Representation Theoretic Notation).

Reviewer #3, comment #9

Figure 1: I believe the use of Figure will help to improve the clarity of the work, but Figure 1 is not very useful. It only helps the reader to understand that the observable and the Hamiltonians are part of the Lie algebra (and it is not that much clear that the star is the observable), which was already clear in the text.

Our response #3.9

We thank the Reviewer for bringing this to our attention. We have added a new figure, which now highlights our main result and hopefully provides more insight to the reader.

Reviewer #3, comment #10

In Equation (5) both norms $\|H\|_K$ and $\|H\|_F$ are used, is it on purpose?

Our response #3.10

Yes, this is intended, but we recognise that this was presented without any introduction to the meaning of the norms. We have moved the main theorems to the Results section, and added further clarification on the norms in Sections 2 C and in the Methods.

Review of representation theory

Reviewer #3, comment #11

In this subsection, the authors present the tools they used from the representation theory in order to help the reader to understand the tools used to obtain their main results. However, I am not sure to who this part is addressed. It is very unclear for people that are not familiar with representation theory, and too many concepts and notions are called but not explained. I will advise to give a more low-level presentation of the tools they use and to give in appendix an extended version of this review.

Our response #3.11

We agree with the Reviewer that it was not clear who this section was intended for. We have now replaced it with Section 2 C (Representation Theoretic Notation) and we explicitly state that it assumes basic knowledge on Lie groups and representation theory. We have also checked that all notation is now properly defined and simplified. Notably, the new version of the paper is significantly more high-level, and therefore only basic knowledge is required to understand the main text.

To accommodate a wider audience, we have also added a low level introduction to the subject of representation theory in the Supplementary Information. This section also now contains all the proofs and discussions we deemed excessively complicated.

Reviewer #3, comment #12

"Consider a complex, finite-dimensional unitary representation $\phi : G \rightarrow U(V_\phi)$: not very clear as V_ϕ not defined (is it the reduced Hilbert space?). Same for Tr_ϕ, U_g .

Our response #3.12

We thank the Reviewer for bring the omission to our attention. We have now added the missing definitions. In addition, we have decided to remove the subscripts under Tr as they are unnecessary.

Reviewer #3, comment #13

The motivation of giving Equation (13) is unclear. This equation is used in the Section IV-A, but it is not clear why it is define and useful.

Our response #3.13

Equation (13) is the definition of I_ϕ , and we recognise that this is poorly explained in the text. It is however an important equation since I_ϕ appears frequently in the proofs. In the new version the equation is in Section 2 C (Representation Theoretic Notation), and therefore the context now makes it clear that it is a definition of a useful quantity. The index now also appears explicitly in our discussion of the main result in Section 2 E (Interpretation of Results), as well as in the proofs in the Methods.

Reviewer #3, comment #14

It is puzzling that the normalized split Casimir invariant (or operator) is presented and motivated in a review and that the authors advise the reader to compare it with "the usual Casimir operator". If the goal of this section is to introduce basic concepts and mathematical tools, it should not rely on a deep knowledge from the reader (or it should be placed in appendix).

Our response #3.14

We agree with the Reviewer that this remark is potentially confusing. We have removed this remark. The (split) Casimir is now only defined in the Methods as it used for the technical lemmas.

We acknowledge that the difference between the split and 'usual' Casimir is not that significant. The discussion on this has now been deferred to the "Introduction to Lie Groups and Representation Theory" section in the Supplementary Information.

Reviewer #3, comment #15

Definition II.1: "where $U_{g_\pm} = \phi(g_\pm)$ for arbitrary $g^+, g^- \in G$. It seems like g^+, g^- are arbitrary elements of G but equation (24) and the following definition of $U(\theta)$ seems to indicate that the signs \pm indicate part of the circuits. It should be clarify in the Definition II.1.

Our response #3.15

We thank the Reviewer for pointing out this unclear passage. The notion has been updated to clarify the difference. Please see the discussion near Definition 2.5.

Reviewer #3, comment #16

This part is essential for the comprehension of the rest of the paper, and we acknowledge that it must be hard to balance between the strict minimum and a full course on Lie Theory. But at least each term and notation used must be properly define somewhere, even if they are very standard for some mathematicians in this field.

Our response #3.16

Indeed calibrating the amount of explanations of Representation and Group Theory has been one of the most significant challenges in writing the manuscript. In this revised version we took the approach of explaining the bare minimum to understand the results, while retaining mathematical accuracy. While we have defined all quantities, for reasons of space we admittedly left explanations short and assumed some previous experience with the concepts. However, we did include a more thorough explanation of the mathematics in the Supplementary Information, in case the reader is curious about particular concepts and wants to know the role they play in the proofs.

Representation-theoretic framework for barren plateaus**Reviewer #3, comment #17**

Theorem III.2: comments on the importance of the terms $\|H\|_K^2$, $\|O\|_K^2$, and $\|P_{\mathfrak{g}}\rho\|_K^2$ would be appreciated. In other work (arXiv:2309.09342) the notion of purity of hermitian operator is introduced and discuss, and I think it is important to also discuss it here. It is discussed in subsections A and B but it could be helpful to have already some comments and a link with this subsection

Our response #3.17

We thank the Reviewer for this suggestion. Indeed the interpretation of the norms is key to interpreting the results. We now included an additional Section 2 E (Interpretation of Results). We have also acknowledged the connection to generalized purity.

Reviewer #3, comment #18

Definition III.3: to improve the clarity, the authors should make the link with other works using reducible Lie Algebra (arXiv:2105.14377, arXiv:2202.00054, ...) and explained again the link with symmetries and subspace preserving ansatzes.

Our response #3.18

We have now added a discussion after Definition 2.7 (Reductive Lie algebra) on the connection with previous notions of symmetric ansatz, which we then further expand in Section 2 G (Comparison with previous approaches).

Reviewer #3, comment #19

Theorem III.5: this result is only introduced by the paragraph just before. It is important to add comments on how it affect quantum ansatzes using reducible Lie Algebra in practice.

Our response #3.19

Following this helpful suggestion by the Reviewer, we have significantly expanded the discussion of the implications of both theorems. This now appears in Section 2 E (Interpretation of Results).

Reviewer #3, comment #20

Theorem III.6: here again, the impact of this theorem is not discussed directly. In addition, the statement of the Theorem should be straightforward "consider a vector $|v\rangle$ in an irrep V " is not clear for example.

Our response #3.20

We agree with the Reviewer that this Theorem is neither clearly stated nor appropriately discussed. We have decided to move this result to the Supplementary Information, Section ‘Projected Norm Lower Bound’ as it requires significantly more prior knowledge to interpret than the other results. In the new section, we have significantly expanded the discussion on its impact.

Reviewer #3, comment #21

”C. Connection with existing approach”: First some terms must be defined “ ϕ is faithful”, or “ ϕ is not faithful” (it was already mentioned but not defined). The connection with the setting of considering subrepresentations is not very clear, especially this sentence” While it may seem that the if the initial state has support across multiple subrepresentations that the variance splits into a sum across the subrepresentations, this is not the case. This is simply because of covariance terms that are not zero in general, i.e. the Haar measure does not always factor across subrepresentations”.

Our response #3.21

We apologize for the missing definitions. We have now defined the notion of a faithful representation.

The comparison with previous approaches was poorly phrased, the discussion has been expanded and clarified in Section 2 G (Comparison with Previous Approaches).

Examples

Reviewer #3, comment #22

We don’t think it is a good idea to present the case uncontrollable systems as just examples. One of the main question before this article was to understand the scaling of the gradient when considering non controllable systems. It should be mentioned with the main theorems.

Our response #3.22

We agree with the Reviewer. All results on the compound ansatz have been promoted to theorems and are now part of the Results section (Section 2 F).

Reviewer #3, comment #23

In addition, the authors should take the time to explained what happens when considering a circuit, with its corresponding Lie Group (that could be $su(d)$), but which is not parametrized enough to reach all unitaries in the Lie Group. This is especially important to avoid any confusion on the meaning of “full controllability”.

Our response #3.23

This is an acute observation by the Reviewer. The setting we have considered is when the ansatz forms an approximate 2-design. However, this does not imply that the ansatz can enact any unitary in its dynamical group, it is in fact a weaker condition, but it is sufficient for all of our results to apply. However, if the circuit is constrained enough so that the 2-design assumption does not hold, then our results would not be valid.

There is now an expanded discussion on the 2-design assumption after Definition 2.5. We emphasize that is an essential assumption and without it none of our results apply. Nonetheless, the new Theorem 2.6 shows that approximate 2-designs can be efficiently realized in realistic settings. Still, we note that different initialization schemes and parameter constraints may lead to different behaviours. We also properly define the notion of controllability in Section 2 G in order to further avoid any confusion.

Reviewer #3, comment #24

We insist: RBS based circuits are NOT always compound ansatz. Only FBS are. And RBS = FBS only when qubits are connected to their nearest neighbors (arXiv:2202.00054, arXiv:2209.08167 & arXiv:2309.15547).

Our response #3.24

This is a correct statement and we apologize for the confusion. We are now explicitly referring to FBS

gates and we removed the mention of RBS gates. We make clear that compound ansatz uses only FBS gates and we are not considering general RBS circuits with arbitrary topology.

General observables

Reviewer #3, comment #25

This section does not have a clear introduction. The authors explained that “we need to allow for observables that have support outside the Lie algebra”, but the content of this section should be summarize at the beginning. A good summary is done at the very last paragraph, but the link with the rest of the section is unclear. It is also very hard to understand Theorem V.1 and the link with the conclusion.

Our response #3.25

We acknowledge that this section as currently written does not connect well with the rest of the paper and can be confusing. We have significantly expanded the discussion around general observables and taken the time to improve the rigor. However, we believe that it is too technical for the general reader and have thus moved it to the Supplementary Information. The main paper now solely focuses around the DLA-BP connection for the LASA case.

REVIEWER COMMENTS

Reviewer #1 (Remarks to the Author):

We thank the authors for their elaborations and corrections; all of the comments I had previously made have now been addressed.

However, I have a concern about a recently added claim to the manuscript, associated with a response to another reviewer (Reviewer #2, authors' Response #2.7). The authors claim that Reviewer #2's Refs. [1-2] only hold when the state lies in an invariant subspace. However, this is not true; rather, Refs. [1-2] just require the ansatz and observable share the same symmetries (i.e., they also work in the "Heisenberg picture" in the authors' terminology). Because of this, I do not believe the authors properly addressed this point.

As an example, Ref. [1] demonstrates how classical simulation of symmetric variational quantum algorithms can be performed even when your initial state does not obey the given symmetries. They achieve this by taking a certain kind of classical shadow of the state; this is generally efficient given access to a quantum computer, but in the case of permutation invariance this can also be done classically given an MPS description of the initial state (which, we emphasize again, does **not** have to satisfy the permutation invariance symmetry). This means the authors' new sentence that "...there have only been a few cases in which potentially useful symmetries, mostly in the Schrodinger picture, have been identified..." on the first page of the manuscript is misleading, and both Refs. [1-2] that Reviewer #2 mentioned should probably be cited here as prior examples where symmetries in the Heisenberg picture were studied.

Reviewer #2 (Remarks to the Author):

In this second round of reviews, I notice that there are many welcome changes to the main draft. Nonetheless, I should qualify that in this round of reviews, I did not have the time to check proofs and go through the appendix in detail especially given the many changes made. By my estimate, roughly a third of the main paper is new material.

Overall, my opinion of the work is similar to as before. This paper formally proves extensions of barren plateaus to the dynamical Lie algebra setting. This will be useful to people studying quantum variational algorithms specifically. Furthermore, the paper is written in a clean way including a nice appendix with background into representation theory. Nonetheless, stepping back from this particular topic, barren plateaus are only one part of the overall story and only one piece of the larger viability issue for quantum variational algorithms. Additionally, finding symmetric problems that fall within this framework is a challenge and often implementing the formulas and ideas in this work can be cumbersome (e.g. many of the norms can be challenging to calculate in general).

Whether this paper meets the criteria of Nature Communications is up to the editor. I would be happy to support the editors' decision either way.

As mentioned earlier, a number of changes were made in this draft. Perhaps the most notable addition is that of approximate convergence to t -designs. These added results are largely a corollary of existing results in the literature. Nonetheless, placed in the context of this paper, I think it adds to the paper nicely. One comment though: I wish the authors were more precise in theorem 2.6 as opposed to just saying $\text{poly}(n)$ especially since I think the dependence on depth is linear (so weakly polynomial). In fact, it seems from their referenced paper (ref [33]) that approximate t -designs arise in linear depth in n , which is also the expected depth from my understanding. As an additional point that may be a point of my own misunderstanding, it also seems that there should be some dependence of the convergence on the particular choice of H_1, \dots, H_d . Perhaps this is hidden in the big O notation.

Other comments:

- This may be my misunderstanding, but in Theorems 2.8 and 2.9, I feel the norm on H can be removed since H_i are a basis and this norm can be arbitrarily scaled by scaling the basis H_i by a constant. I do not believe this norm has any physical meaning? Perhaps the authors should comment on this and note whether there is a canonical basis where this norm is always equal to one. This may simplify the theorems a bit too.
- A table of contents would be useful for the appendix.

- The authors discuss classical simulability in the discussion. A few references are listed below in this regard which I think are instances of classical simulability of symmetric quantum systems and which I think the authors could reference and potentially discuss further.
- A times the authors write statements such as "If the DLA has polynomial dimension and the generators are sums of Pauli strings, then there is an efficient procedure for discovering the simple ideals. Given a basis..." Statements like this seem to be too general and say that it is easy to find the DLA, but I am not so convinced. For example, in a few places, the authors assume a basis is known, and even though finding such a basis may be a polynomial time operation, this polynomial time could be prohibitively expensive. Also, many times, the symmetries of a problem are not apparent in the way the problem is given. I think the authors should qualify this a bit when they write such things or be more explicit about runtimes and input/outputs.

References:

- [1] Terhal, Barbara M., and David P. DiVincenzo. "Classical simulation of noninteracting-fermion quantum circuits." *Physical Review A* 65.3 (2002): 032325.
- [2] Anschuetz, Eric R., et al. "Efficient classical algorithms for simulating symmetric quantum systems." *Quantum* 7 (2023): 1189.
- [3] Somma, Rolando, et al. "Efficient solvability of Hamiltonians and limits on the power of some quantum computational models." *Physical review letters* 97.19 (2006): 190501.

Reviewer #3 (Remarks to the Author):

We are satisfied with the changes made by the authors.

We therefore recommend publication of this paper.

Response to the Reviewers

Re: Manuscript ID NCOMMS-23-49122

“The Adjoint Is All You Need:
Characterizing Barren Plateaus in Quantum Ansätze”

Enrico Fontana, Dylan Herman, Shouvanik Chakrabarti, Niraj Kumar, Romina Yalovetzky, Jamie Heredge,
Shree Hari Sureshbabu, Marco Pistoia

Dear Reviewers,

We would like to thank you for your assessment of our work and providing critical feedback, which we have used to improve the presentation. Enclosed we submit our revised manuscript. The changes to the manuscript that directly answer comments or we would like you draw attention to are indicated in **blue font**. Below we address the comments individually.

Thank you,

Enrico Fontana on behalf of all authors.

Reviewer #1, comment #1

We thank the authors for their elaborations and corrections; all of the comments I had previously made have now been addressed.

Our response #1.1

We are delighted to hear that the previous corrections have been satisfactory to the Reviewer.

Reviewer #1, comment #2

*However, I have a concern about a recently added claim to the manuscript, associated with a response to another Reviewer (Reviewer #2, authors' Response #2.7). The authors claim that Reviewer #2's Refs. [1-2] only hold when the state lies in an invariant subspace. However, this is not true; rather, Refs. [1-2] just require the ansatz and observable share the same symmetries (i.e., they also work in the "Heisenberg picture" in the authors' terminology). Because of this, I do not believe the authors properly addressed this point. As an example, Ref. [1] demonstrates how classical simulation of symmetric variational quantum algorithms can be performed even when your initial state does not obey the given symmetries. They achieve this by taking a certain kind of classical shadow of the state; this is generally efficient given access to a quantum computer, but in the case of permutation invariance this can also be done classically given an MPS description of the initial state (which, we emphasize again, does *not* have to satisfy the permutation invariance symmetry). This means the authors' new sentence that "...there have only been a few cases in which potentially useful symmetries, mostly in the Schrodinger picture, have been identified..." on the first page of the manuscript is misleading, and both Refs. [1-2] that Reviewer #2 mentioned should probably be cited here as prior examples where symmetries in the Heisenberg picture were studied.*

*[1] Terhal, Barbara M., and David P. DiVincenzo. "Classical simulation of noninteracting-fermion quantum circuits." *Physical Review A* 65.3 (2002): 032325. [2] Anschuetz, Eric R., et al. "Efficient classical algorithms for simulating symmetric quantum systems." *Quantum* 7 (2023): 1189.*

Our response #1.2

We agree with the Reviewer that the added statement is misleading, and we apologize for the inaccuracy. By "use of symmetries", we meant specifically in the context of analyzing barren plateaus. To emphasize this distinction we have now changed the sentence. We have also added a new sentence thereafter to clarify that indeed symmetries have been studied before mainly in the context of classical simulation, as correctly pointed out by both Reviewers #1 and #2.

The Reviewer is also correct in noting that Heisenberg symmetries have been used before for simulation, therefore we have added Ref. [1] as well as some other. However, based on our understanding, Ref. [2] (Anschuetz *et al.*) still utilizes symmetries in the Schrödinger picture. Specifically, the simulation is done separately in each invariant subspace of quantum states by expressing the input state in the Schur basis, as opposed to using invariant subspaces of observables. The latter is what we mean by Heisenberg picture, the former being Schrödinger. We now explicitly define what we mean by these terms at the end of Section 2C.

We agree that the initial state does not have to respect permutation invariance in Ref. [2]. We added a statement to the Introduction to indicate that the symmetry enables separate simulation on each of the invariant subspaces. Specifically:

“Previously, symmetries have been leveraged for efficient classical simulation of quantum circuits in both the Schrödinger [2] and Heisenberg [1, others] pictures. The simulation is performed separately in each invariant subspace defined by the symmetries by projecting the states or operators accordingly.”

Reviewer #2, comment #1

In this second round of reviews, I notice that there are many welcome changes to the main draft. Nonetheless, I should qualify that in this round of reviews, I did not have the time to check proofs and go through the appendix in detail especially given the many changes made. By my estimate, roughly a third of the main paper is new material.

Our response #2.1

We understand that the paper may look very different. We apologize for the many changes, which were largely the consequence of adapting the paper to the format of Nature Communications and trying to make the text appeal to a broader audience, as well as addressing the Reviewers’ replies. We remark that effectively the only new results are the improved bound on the mixing time and extra details on detecting symmetries in arbitrary circuits. The former, we derived in the meanwhile and thought significant enough to include since it strengthens the paper considerable and addresses multiple comments by the Reviewers. The latter also addresses comments by the Reviewers and makes the paper stronger. We hope the Reviewer understands and forgives the different look of the paper.

Reviewer #2, comment #2

Overall, my opinion of the work is similar to as before. This paper formally proves extensions of barren plateaus to the dynamical lie algebra setting. This will be useful to people studying quantum variational algorithms specifically. Furthermore, the paper is written in a clean way including a nice appendix with background into representation theory. Nonetheless, stepping back from this particular topic, barren plateaus are only one part of the overall story and only one piece of the larger viability issue for quantum variational algorithms. Additionally, finding symmetric problems that fall within this framework is a challenge and often implementing the formulas and ideas in this work can be cumbersome (e.g. many of the norms can be challenging to calculate in general).

Our response #2.2

We thank the Reviewer for the positive comments. Regarding the pitfall identified, we do agree with the Reviewer that barren plateaus are only one of many issues with variational quantum algorithms, which we highlight in the introduction and discussion. However, the BPs problem has in recent years received significant attention and is frequently cited as one of the major bottlenecks for variational algorithms, see for instance “Variational quantum algorithms”, Cerezo *et al.*. BPs are important to consider when attempting to train a variational algorithm, regardless if convergence to quality optima is possible, and thus absence of BPs is a necessary condition to successful training. In addition, before recent advances BPs were largely considered intractable beyond a handful of special cases. Therefore, we believe the results outlined in the manuscript have a potential for impact that compensates for their unarguable specificity.

It is also conjectured that the DLA dimension controls the presence or absence of some of the other known issues with VQAs, e.g. in “Theory of overparametrization in quantum neural networks”, Larocca *et al.* showed evidence that when the number of parameters scales with the DLA dimension convergence

to a high quality solution becomes possible. This is another potential future application of our techniques. In another work, “A Convergence Theory for Over-parameterized Variational Quantum Eigensolvers” by You *et al.* showed that for $SU(2^n)$ the stronger condition of linear convergence to the global minimum becomes possible when the number of parameters reaches a critical threshold, appearing to be polynomial in DLA dimension. Importantly, the ability to compute second Haar moments appears in their analysis, which is where the DLA dimension factors in. Given that our results present techniques for calculating these integrals in the case of a more general ansatz, i.e. all LASAs, they may enable one to derive a more general theory of convergence for VQAs. We had noted this in the Discussion, saying that our techniques can help analyze the projected gradient flow dynamics that occurs in the LASA setting. Specifically, we can now analyze dynamics where the tangent space is not the well-understood $\mathfrak{su}(2^n)$.

Regarding symmetries, we emphasize that the only condition for our results to hold is the LASA assumption, i.e. the observable is in the DLA. This by default is satisfied by any quantum approximate optimization/alternating operator ansatz (QAOA), Hamiltonian variational ansatz, and some equivariant quantum neural networks. Thus, we believe this requirement is commonly satisfied. Still, even though symmetries are commonplace, there may be an issue in characterizing which symmetry is present, i.e. the isomorphism classes of the simple ideals of the DLA. We give some indications on how to at least determine a DLA basis in the paper, in Section 2E, which we have enhanced in the revised version. Full classification is believed to be a hard problem, even though progress has been made recently in “Classification of dynamical Lie algebras for translation-invariant 2-local spin systems in one dimension” by Wiersema *et al.* We acknowledge that the norms can be challenging to analyze in general. However, in the polynomial DLA setting it is at least possible, with the procedure we highlighted in the main text (Section 2E), to get an idea of the scaling from small experiments.

Overall, we would like to note that the intention of the paper was to provide a natural explanation for an observed phenomenon, the correlation between DLA dimension and BPs, and as such improve our understanding of BPs in the LASA case and in general. While occasionally possible, the ability of our model to exactly calculate cost and gradient variances may not in itself always provide a useful method for predicting BPs, especially where the symmetry is unknown beforehand and must be inferred from experiment. Naturally, if one needs to simulate or run quantum circuits to infer the symmetry, it would make more sense to just calculate the variances directly. The methods still have a very useful role where the symmetries can be characterised algebraically (like the quantum compound ansatz), and these settings are rapidly growing in number thanks to rapid theoretical advances. However, in our view the major contribution is still the conceptual one of being able to give a clear, concise, and mathematically exact answer to a phenomenon that had resisted theoretical understanding and that is commonplace in variational quantum algorithms. We have added clarifications to the manuscript’s Section 2E around this point.

Reviewer #2, comment #3

As mentioned earlier, a number of changes were made in this draft. Perhaps the most notable addition is that of approximate convergence to t -designs. These added results are largely a corollary of existing results in the literature. Nonetheless, placed in the context of this paper, I think it adds to the paper nicely. One comment though: I wish the authors were more precise in theorem 2.6 as opposed to just saying $\text{poly}(n)$ especially since I think the dependence on depth is linear (so weakly polynomial). In fact, it seems from their referenced paper (ref [33]) that approximate t -designs arise in linear depth in n , which is also the expected depth from my understanding. As an additional point that may be a point of my own misunderstanding, it also seems that there should be some dependence of the convergence on the particular choice of H_1, \dots, H_d . Perhaps this is hidden in the big O notation.

Our response #2.3

We appreciate the Reviewer bringing up the comparison of our results to the linear depth scaling of Ref. [33]. The result we present only focuses on lower bounding the spectral gap and hence the number of evolutions $e^{-\theta_i H_k}$ that need to be applied for the walk to mix. This is what we mean by layers. The overall depth of the circuit can be significantly larger in terms of two-qubit and one-qubits. We now emphasize this in the manuscript. Note that the depth scaling in Ref. [33] for implementing each evolution (multi-qubit Pauli rotations) is actually $\log(n)$, defined as the additional multiplicative factor d in the abstract of Ref. [33]. The additional linear scaling of n in Ref. [33], comes from applying derandomization (using “Explicit

orthogonal and unitary designs” O’Donnell *et al.*) and corresponds to the random seed length to be passed to a pseudorandom number generator. We assume to be in the truly random setting, and thus only focus on spectral gap and mixing time. This assumption has been made before in literature, for example in “Local random quantum circuits are approximate polynomial-designs” by Brandão *et al.*. Note that in Ref. [33], they obtain a lower bound of $\Omega(1/t)$ on the spectral gap leading to $O(t \log(1/\epsilon))$ layers or random walk steps, which ignores the factor of $\log(n)$ corresponding to the depth required to implement each Pauli rotation, and assumes the truly random setting (hence no n factor). We now highlight this comparison in the main text right after Theorem 2.6 is presented.

The more general result in the Supplementary Information Section 7 Theorem 6 also highlights that for arbitrary DLA the layer scaling depends on a quantity we call the stable Killing rank, r_K . Specifically, it depends on the ratio d_g/r_K , where r_K is the minimum stable Killing rank for the chosen basis. Specifically, the spectral gap bound of Ref. [33] shows that $d_g/r_K = \mathcal{O}(1)$ for Pauli rotations and $SU(2^n)$. The overall depth scaling in terms of one and two qubit gates includes additional factors of n from derandomization and $\log(n)$ for implementing the rotations, as noted in the previous paragraph. For arbitrary DLAs and generators, it is unclear if $d_g/r_K = \mathcal{O}(1)$ is possible. For example, Theorem 2.10 shows that for compound layers and the basis of Jordan-Wigner encoded qudit Paulis, $d_g/r_K = n$, an additional factor of n over the $SU(2^n)$ Pauli rotation setting. We do show in the appendix that $r_K = \Omega(\sqrt{d_g})$, so if $d_g = O(\text{poly}(n))$, the most we can say is that $d_g/r_K = O(\text{poly}(n))$. There is definitely room for further improvement in future work.

The Reviewer is correct that in general there is an additional dependence on spectral quantities related to H_i . However, there is no explicit dependence on the norms. This is because the θ s are chosen uniformly from the period of the generator, which can compensate for the small norm. The spectral quantity that does appear and depends on the basis is the ratio d_g/r_K mentioned earlier. r_K is the ratio of the Frobenius norm of ad_{iH} to its spectral norm, and hence a continuous, rank-like quantity. Note that the dependence on these spectral quantities is not hidden in the big O for Theorem 2.6, as mentioned $d_g/r_K = O(\text{poly}(n))$ for poly DLA.

For readability purposes and to appeal to a broader audience, we felt that it was better to keep the simpler corollary for polynomial DLA in the main text only and leave the more general result (Supplementary Theorem 6) discussing the Killing rank to the supplementary materials. We are happy to add the more general result to the main text if the Reviewer feels that it will reduce some confusions.

Reviewer #2, comment #4

This may be my misunderstanding, but in Theorems 2.8 and 2.9, I feel the norm on H can be removed since H_i are a basis and this norm can be arbitrarily scaled by scaling the basis H_i by a constant. I do not believe this norm has any physical meaning? Perhaps the authors should comment on this and note whether there is a canonical basis where this norm is always equal to one. This may simplify the theorems a bit too.

Our response #2.4

We appreciate the Reviewer bringing this to our attention. We believe the confusion might stem from the use of H for both basis elements, i.e. in Theorem 2.6, and an arbitrary skew-hermitian generator in the DLA. We have changed Theorem 2.6 to use different notation. Now, in the main text H refers to an arbitrary DLA element. In Theorems 2.8 and 2.9, H is meant to be an arbitrary generator (not necessarily a basis element), like O is an arbitrary observable in the DLA. Even if H is set up to be a basis element, the norm of the generator used may not be 1 (Theorem 2.6 doesn’t require a normalized basis, just orthogonal). For example, in the case of compound layers (Theorem 2.11), the Killing norms of the generators are not 1, they have Killing norm around $n/2$ for n qubits.

As shown in Equation 12, the gradient depends on the generator H of the gate whose gradient is computed, i.e. $e^{-\theta H}$, and hence there is a spectral dependence on H . The (Killing) norm on H is the Frobenius norm of the operator ad_H , there could be a generator such that this norm is one, but we believe that this is the same as saying, we could have an observable O whose standard Frobenius norm is 1. We could always scale H and O to have their norms be 1, but then our ansatz would have changed. Overall it does not appear to us that there should be a fixed basis introduced into these basis independent expressions.

Reviewer #2, comment #5

A table of contents would be useful for the appendix.

Our response #2.5

We thank the Reviewer for the suggestion. We have added a table of contents to the appendix.

Reviewer #2, comment #6

The authors discuss classical simulability in the discussion. A few references are listed below in this regard which I think are instances of classical simulability of symmetric quantum systems and which I think the authors could reference and potentially discuss further.

*References: [1] Terhal, Barbara M., and David P. DiVincenzo. "Classical simulation of noninteracting-fermion quantum circuits." *Physical Review A* 65.3 (2002): 032325. [2] Anschuetz, Eric R., et al. "Efficient classical algorithms for simulating symmetric quantum systems." *Quantum* 7 (2023): 1189. [3] Somma, Rolando, et al. "Efficient solvability of Hamiltonians and limits on the power of some quantum computational models." *Physical review letters* 97.19 (2006): 190501.*

Our response #2.6

We thank the Reviewer for the helpful suggestion. We have added the mentioned references to the Discussion. Following a similar suggestion by Reviewer #1, we also reference them in the Introduction, with some small discussion on the connection to symmetric quantum systems.

We remark that the simulation technique that most strongly connects to our results is the g -sim technique ("Lie-algebraic classical simulations for variational quantum computing" Goh *et al.*), as mentioned in the Discussion section. This is mainly because of the explicit dependence of this simulation technique on the dimension of the DLA. Other simulation algorithms for symmetric systems are probably related but, while definitely interesting, a complete analysis of the interplay of BPs, symmetries and simulatability is beyond the scope of our paper. For this reason in the Discussion we limit ourselves to mentioning that in the future one may explore vanishing gradients in other symmetric simulatable systems.

Reviewer #2, comment #7

A times the authors write statements such as "If the DLA has polynomial dimension and the generators are sums of Pauli strings, then there is an efficient procedure for discovering the simple ideals. Given a basis. . ." Statements like this seem to be too general and say that it is easy to find the DLA, but I am not so convinced. For example, in a few places, the authors assume a basis is known, and even though finding such a basis may be a polynomial time operation, this polynomial time could be prohibitively expensive. Also, many times, the symmetries of a problem are not apparent in the way the problem is given. I think the authors should qualify this a bit when they write such things or be more explicit about runtimes and input/outputs.

Our response #2.7

The Reviewer is correct in pointing out that the statements about an efficient procedure for discovering the ideals were poorly motivated. We have attempted to remedy this by expanding the details of the procedure. We now are more explicit on the scaling of this procedure and on its limitations, in particular we now take into account the potential blow-up of the DLA basis vectors when expressed as Pauli operators.

We are confident the procedure could be improved upon and made into a viable method; however with these paragraphs we simply intended to show that it may be possible in some cases to identify the ideals without a quantum computer. While a universal and efficient method would be extremely valuable, it would also be orthogonal to the message of the paper, which instead focuses on providing an exact theoretical explanation for an observed phenomenon, that of BPs in symmetric ansatzes. Fully exploring the predictive potential of our results may yet necessitate significant theoretical developments beyond those presented in the manuscript. Still we note that the theory is readily applicable as a predictive tool when the symmetries are well characterized (as demonstrated for quantum compound ansatz), and will necessarily form the foundation of any future method to determine BPs in arbitrary settings.

REVIEWERS' COMMENTS

Reviewer #1 (Remarks to the Author):

We thank the authors for their elaborations; all of the comments I had previously made have now been addressed.